# Training with Mixed-Precision Floating-Point Assignments

**Wonyeol Lee**                                                    *wonyeol.lee.cs@gmail.com*
*Stanford University, USA*

**Rahul Sharma**                                                   *rahsha@microsoft.edu*
*Microsoft Research, India*

**Alex Aiken**                                                     *aaiken@stanford.edu*
*Stanford University, USA*

**Reviewed on OpenReview:** *https: // openreview. net/ forum? id= ZoXi7n54OB*

## Abstract

When training deep neural networks, keeping all tensors in high precision (e.g., 32-bit or even 16-bit floats) is often wasteful. However, keeping all tensors in low precision (e.g., 8-bit floats) can lead to unacceptable accuracy loss. Hence, it is important to use a precision assignment—a mapping from all tensors (arising in training) to precision levels (high or low)—that keeps most of the tensors in low precision and leads to sufficiently accurate models. We provide a technique that explores this memory-accuracy tradeoff by generating precision assignments for convolutional neural networks that (i) use less memory and (ii) lead to more accurate convolutional networks at the same time, compared to the precision assignments considered by prior work in low-precision floating-point training. We evaluate our technique on image classification tasks by training convolutional networks on CIFAR-10, CIFAR-100, and ImageNet. Our method typically provides $> 2\times$ memory reduction over a baseline precision assignment while preserving training accuracy, and gives further reductions by trading off accuracy. Compared to other baselines which sometimes cause training to diverge, our method provides similar or better memory reduction while avoiding divergence.

## 1 Introduction

In deep neural network training, floating-point formats are usually used to represent tensors and it is worthwhile to use the smallest bitwidth format that gives acceptable results. For example, it is common to replace tensors using 32-bit floats with tensors that use 16-bit floats (Kalamkar et al., 2019; Micikevicius et al., 2018). The benefits are easy to understand: computations using lower-precision floats not only use less memory but are also faster (due to improved vector parallelism, locality, and reduced data movement). The downside is that there is generally some loss of training accuracy, and in the worst case training may not even converge.

For such *low-precision floating-point training*, the most common approaches use two floating-point formats—one for lower-precision floats (e.g., 8-bit floats) and the other for higher-precision floats (e.g., 16-bit floats)—and assign one of the two formats to each tensor (including weights, activations, and their gradients). The precision assignments studied in previous work fall into one of two assignment schemes (which both have several variants): the *uniform* assignment uses low precision for almost all tensors (often excepting those in the first and/or last few layers) (Micikevicius et al., 2018), while the *operator-based* assignment limits low precision to the input tensors of certain operators (e.g., convolutions) (Sun et al., 2019). Prior work has shown that both precision assignment schemes (with well-chosen low-bitwidth floating-point formats) can match the accuracy of 32-bit-float training (Cambier et al., 2020; Chmiel et al., 2021; Drumond et al., 2018; Fox et al., 2021; Kalamkar et al., 2019; Micikevicius et al., 2018; Sun et al., 2019; Wang et al., 2018).

There is an important limitation in all prior approaches to low-precision floating-point training: they use very few precision assignments (most often just one) for a given set of models, but there are some other

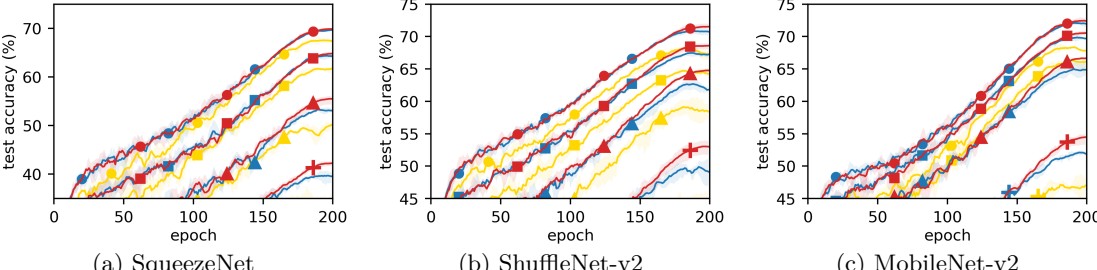

(a) SqueezeNet        (b) ShuffleNet-v2        (c) MobileNet-v2

Figure 1: Training trajectory of various models on CIFAR-100. Colors denote precision assignments: all-32-bit $\pi_{\mathrm{fp32}}$ (red), uniform $\pi_{\mathrm{unif}}$ (yellow), and operator-based $\pi_{\mathrm{op}}$ (blue) (see §3.1); the latter two use the 8-bit (and 16-bit) floats in Sun et al. (2019) as low (and high) precision numbers. Markers denote the "width multiplier" of a model, which controls the capacity of the model (see §5.3): 1.0 (●), 0.5 (■), 0.25 (▲), and 0.1 (✚). Some lines of $\pi_{\mathrm{unif}}$ are missing as they converge to small values or diverge. Observe that neither $\pi_{\mathrm{unif}}$ nor $\pi_{\mathrm{op}}$ works best for all models: in some models, $\pi_{\mathrm{op}}$ has a similar accuracy to $\pi_{\mathrm{fp32}}$; but in other (and all) models, the accuracy drop of $\pi_{\mathrm{op}}$ (and $\pi_{\mathrm{unif}}$) from $\pi_{\mathrm{fp32}}$ are noticeably large (i.e., $>1\%$).

models and inputs where the chosen precision assignment (i) results in noticeably worse accuracy than 32-bit-float training, (ii) causes training to even diverge, or (iii) admits a more efficient assignment that achieves similar training accuracy (see Figures 1, 3, and 4).

In this paper, we present a new, automated method for choosing precision assignments that removes the limitations described above. To do so, we formally introduce the *memory-accuracy tradeoff problem* (§3.2): given a dataset, a model, and two floating-point precision levels (i.e., bitwidths; high and low), find a *mixed precision assignment* (a mapping from all tensors arising in training to high/low precision) for the model that maximizes training accuracy subject to a given upper bound on the *model aggregate* (i.e., the total number of bits of all tensors appearing in training). The model aggregate is a proxy for the memory and time required for training, as it is roughly proportional to memory footprint and also well-correlated with training time (since training is often dominated by data movement) (Micikevicius et al., 2018).

We prove that the memory-accuracy tradeoff problem is theoretically difficult (namely NP-hard) partly due to the exponential number of possible mixed precision assignments (which we often refer to simply as precision assignments for brevity) (§3.3). The large number of possible assignments makes the problem difficult in practice as well: there is no known analytical method for predicting the training accuracy of a given precision assignment, and for any practical model there are far too many precision assignments to simply test them all.

We propose a simple (heuristic) approach to the tradeoff problem that prioritizes tensors for low-precision formats based on the tensor's size (with an additional step described below) (§4.1). More specifically, our algorithm takes as input a single parameter giving a desired upper bound on the model aggregate. Starting with the largest tensor in the model, tensors are assigned low precision in size order (from largest to smallest) until the model aggregate falls below the given upper bound; all remaining tensors are assigned high precision. Our main result is that this method discovers mixed precision assignments that use less memory while achieving higher training accuracy than previous approaches. While we cannot show that our method finds Pareto-optimal memory-accuracy tradeoffs, we do show that our results are closer to Pareto-optimal than prior methods.

Some precision assignments initially generated by our algorithm cause training to diverge due to an excessive number of overflows. To address this issue, we propose an overflow handling technique that promotes tensors causing too many overflows from low precision to high precision during training (§4.2). In our experiments, these promotions consume only a small amount of additional memory ($<3\%$ of the maximum model aggregate) and prevent training from diverging. The overflow handling technique is not specific to our algorithm and can be applied to other precision assignment methods as well.

We evaluate a PyTorch implementation of our method on standard image classification tasks by training four convolutional networks (and their variants) on CIFAR-10, CIFAR-100, and ImageNet (§5). We first demonstrate that the precision assignments computed by our method alleviate the limitations of existing methods: they indeed explore the tradeoff between memory and accuracy and exhibit a better tradeoff than

the uniform and operator-based assignments. We then show the two main components of our method (i.e., precision demotion of larger tensors and precision promotion of overflowing tensors) are both important to produce competitive precision assignments. We also provide some guidance on how users may apply our method to navigate the memory-accuracy tradeoff.

To summarize, this work makes four main contributions:

- We formally introduce the memory-accuracy tradeoff problem to explore better mixed precision assignments for low-precision floating-point training and prove the NP-hardness of the problem.
- We present a novel precision assignment technique, as a heuristic solution to the tradeoff problem, that proposes assignments based on a single parameter denoting a desired upper bound on the model aggregate.
- We present a novel technique that handles an excessive number of overflows arising in training while using a small amount of additional memory. The technique is applicable to any (not just our) precision assignments.
- We demonstrate that the mixed precision assignments found by our method do explore the tradeoff between memory and training accuracy, and outperform existing precision assignment methods.

We remark that this work focuses on low-precision *floating-point* training, not *fixed-point* training (which uses fixed-point formats), since we want to target upcoming (or very recent) hardware with native support for low-precision floats (e.g., 8-bit floats) and their operations (e.g., Andersch et al. (2022)). Also, this work focuses on low-precision *training* (which trains a model from scratch), not *inference* (which assumes a pre-trained model). More discussion is in §2. We further remark that in our experiments we simulate low-precision formats (e.g., 8-bit floats) with 32-bit floats as in prior works, since a hardware and software ecosystem that natively supports these formats does not yet exist. Similarly, we do not include other models (e.g., language models) in the experiments, since no current software stacks support per-tensor precision assignments for certain operators that those models use. More details are in §5.1 and §5.2.

For image classification tasks and convolutional networks, our precision assignment method typically provides $> 2\times$ memory reduction over the operator-based assignment while maintaining similar training accuracy and gives further reductions by trading off accuracy. Our method also provides similar memory reduction to the uniform assignment, while avoiding the divergence of training often caused by a uniform assignment.

The paper is organized as follows. After discussing related work (§2), we define the memory-accuracy tradeoff problem and study its hardness (§3). We then describe our algorithm for the problem (§4) and our evaluation (§5). We conclude with limitations and future work (§6).

## 2 Related Work

**Low-precision floating-point training** has been extensively studied since the work of Micikevicius et al. (2018). One active research direction is to select appropriate floating-point formats (or their variants) for low- and high-precision numbers in training. Various floating-point formats have been proposed, including FP16 (Micikevicius et al., 2018), BF16 (Kalamkar et al., 2019), FP8 (Micikevicius et al., 2022; Wang et al., 2018), HFP8 (Sun et al., 2019), and FP6 (Chmiel et al., 2021), along with some variants such as HBFP (Drumond et al., 2018), S2FP8 (Cambier et al., 2020), and BM (Fox et al., 2021). Recently, the problem of automatically selecting such floating-point formats has been considered (Yang et al., 2022). Another research direction is to develop algorithmic techniques that improve training accuracy under low precision: e.g., (Björck et al., 2021; Sa et al., 2018; Yang et al., 2019a; Zamirai et al., 2020). Our work is orthogonal and complementary to all these prior works: they consider various floating-point formats or training algorithms but use a *fixed* precision assignment, which is either the uniform or operator-based assignment (or their variants); our work explores *various* precision assignments once floating-point formats and training algorithms are fixed (e.g., based on the prior works). The tradeoff between memory and accuracy in training is also considered in Yang et al. (2022), but the work differs from ours: they vary *floating-point formats* when a precision assignment is fixed, while we vary *precision assignments* when floating-point formats are fixed.

**Low-precision fixed-point training** uses fixed-point formats as a low-precision representation instead of a floating-point format. Some works use fixed-point formats for forward tensors and floating-point formats for backward tensors: e.g., (Choi et al., 2018; Courbariaux et al., 2015; Jacob et al., 2018; Sun et al., 2020; Yang et al., 2019b). Other works use only fixed-point formats for all tensors: e.g., (Banner et al., 2018; Das et al.,

2018; Gupta et al., 2015; Rajagopal et al., 2020; Sakr & Shanbhag, 2019; Wu et al., 2018; Zhang et al., 2020; Zhou et al., 2016). Among all these works, some consider various mixed precision assignments with different bitwidth (fixed-point) formats: (Sakr & Shanbhag, 2019; Zhang et al., 2020); but they are not applicable to our context (i.e., floating-point training) since they rely on some properties of *fixed-point* formats that do not hold for *floating-point* formats (e.g., all numbers in a given format are equally distributed). The approach taken in Rajagopal et al. (2020) is orthogonal and complementary to ours: they use only the *uniform* precision assignment, but change the underlying low-precision formats during training; we consider various *mixed* precision assignments, but fix the underlying low-precision formats during training.

**Low-precision inference**, often called neural network quantization (in a narrow sense), aims at reducing the latency or memory of neural network inference (instead of training) by using low-precision numbers (Nagel et al., 2021). Existing approaches typically assume a pre-trained model and try to find low-precision formats for each part of the inference computation, either by retraining the model (called quantization-aware training) or without any retraining (called post-training quantization); see, e.g., Gholami et al. (2022); Qin et al. (2022) for surveys. Some works on inference consider various mixed precision assignments, but they are not applicable to our context: they focus on making inference more efficient and usually assume a pre-trained model; we focus on making training more efficient and aim at learning a model from scratch.

**Floating-point tuning** is another related topic, which considers the following problem: given a program, assign appropriate formats (among given candidates) to the program's floating-point variables such that the program's output has an error smaller than a given threshold for all given inputs, while also maximizing performance (Chiang et al., 2017; Guo & Rubio-González, 2018; Menon et al., 2018; Rubio-González et al., 2013; 2016). This problem is different from the problem we focus on: the former considers the *floating-point error* after a single run of a program, while we consider the *training accuracy* after a large number of runs of a program (i.e., a gradient computation) where each run affects the next run; further, the former considers *general-purpose* programs, while we consider *deep learning* programs and exploit their unique features.

## 3 Problem

In this section, we first provide background on low-precision floating-point training (§3.1), based on which the memory-accuracy tradeoff problem is introduced (§3.2). We then prove the NP-hardness of the problem (§3.3). Our approach in §3–4 is more formal than most related works for two reasons: (i) we show the problem is NP-hard, which has not been considered in prior work; and (ii) to clearly describe the precision assignments to be considered.

### 3.1 Background: Low-Precision Floating-Point Training

Let $\mathbb{T}$ be the set of real-valued tensors and let $[n]$ denote the set $\{1, \ldots, n\}$. For a supervised learning task, we usually consider a *model network* $\mathcal{M} = (f_1, \ldots, f_n)$ parameterized by $\boldsymbol{\theta} = (\theta_1, \ldots, \theta_n) \in \mathbb{T}^n$, and a *loss network* $\mathcal{L} = (f_{n+1}, \ldots, f_m)$, where $f_i : \mathbb{T}^2 \to \mathbb{T}$ is a primitive operator on tensors (e.g., convolution, batch normalization, maxpool, and softmax). Given an input-output pair $(x, y) \in \mathbb{T}^2$, the model $\mathcal{M}$ computes a predicted output $y'$ of $x$ by iteratively applying $f_i(\cdot, \theta_i)$ to $x$ ($i \in [n]$), and $\mathcal{L}$ computes a loss from $y'$ by iteratively applying $f_{i'}(\cdot, y)$ to $y'$ ($i' \in [m] \setminus [n]$). A standard way to train $\mathcal{M}$ is to minimize the loss value using the gradient descent algorithm: iteratively update $\boldsymbol{\theta}$ by following the gradient of the loss with respect to $\boldsymbol{\theta}$.

**Floating-point training.** In practice, we perform a gradient computation usually with tensors represented in floating-point formats. Let $\pi : \mathsf{TS} \to \mathsf{FP}$ be a *precision assignment* giving the floating-point format of each tensor, where $\mathsf{TS} \triangleq \{v_i, dv_i, \theta_j, d\theta_j \mid i \in [m+1], j \in [n]\}$ is the set of tensors arising in a gradient computation (explained below), and $\mathsf{FP} \triangleq \{\mathsf{fp}(e, m, b) \mid e, m \in \mathbb{N}, b \in \mathbb{Z}\}$ is the set of floating-point formats. Here $\mathsf{fp}(e, m, b)$ denotes a floating-point format that consists of a 1-bit sign, an $e$-bit exponent, and an $m$-bit mantissa, and has an (additional) exponent bias of $b \in \mathbb{Z}$. A common choice of $\pi$ is $\pi_{\mathrm{fp32}}(t) \triangleq \mathsf{fp32}$ for all $t \in \mathsf{TS}$, where $\mathsf{fp32} \triangleq \mathsf{fp}(8, 23, 0)$ is the standard 32-bit floating-point format.

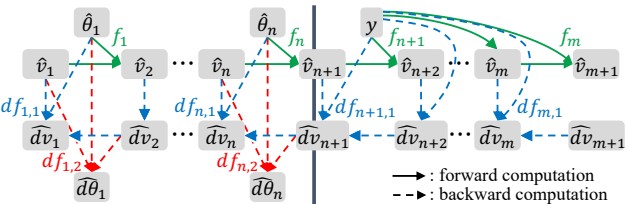

Figure 2: A diagram showing the tensors and operators used in a gradient computation; see Eq. (1) for details. For brevity, rounding functions $\text{rnd}_{\pi(\cdot)}$ are omitted.

Given a precision assignment $\pi$, a gradient computation is typically performed by the backpropagation algorithm: with $\hat{v}_1 = \text{rnd}_{\pi(v_1)}(x)$ and $\hat{dv}_{m+1} = \text{rnd}_{\pi(dv_{m+1})}(1)$, compute

$$\begin{aligned}
\hat{v}_{i+1} &= \text{rnd}_{\pi(v_{i+1})}(f_i(\hat{v}_i, \hat{u}_i)), & \hat{\theta}_j &= \text{rnd}_{\pi(\theta_j)}(\theta_j), \\
\hat{dv}_i &= \text{rnd}_{\pi(dv_i)}(df_{i,1}(\hat{dv}_{i+1}, \hat{v}_i, \hat{u}_i)), & \hat{d\theta}_j &= \text{rnd}_{\pi(d\theta_j)}(df_{j,2}(\hat{dv}_{j+1}, \hat{v}_j, \hat{\theta}_j)),
\end{aligned} \tag{1}$$

for $i \in [m]$ and $j \in [n]$; see Figure 2 for a diagram. Here $\text{rnd} : \text{FP} \times \mathbb{T} \to \mathbb{T}$ is a function rounding a given input to a given floating-point format, $df_{i,1}, df_{i,2} : \mathbb{T}^3 \to \mathbb{T}$ are the backward operators of $f_i$ with respect to its first and second arguments, respectively, and $\hat{u}_i = \hat{\theta}_i$ if $i \in [n]$ and $y$ otherwise. We call $v_i$ and $\theta_j$ the *forward* tensors, and $dv_i$ and $d\theta_j$ the *backward* tensors. We put a hat over each tensor to emphasize that its value is the output of a rounding function to a possibly low-precision format; remark that such a rounding function is not used within $f_i$, $df_{i,1}$, and $df_{i,2}$, since they typically use large bitwidth floats (e.g., fp32) and no low-precision floats internally (Cambier et al., 2020; Kalamkar et al., 2019). After the computation, $\hat{d\theta}_j$ stores the gradient of the loss value with respect to $\theta_j$.

The overall picture of floating-point training is now described as follows. In each iteration of the gradient descent algorithm, we compute $\hat{d\theta}_j$ via Eq. (1) using a given precision assignment $\pi$, training data $(x, y)$, and current weights $\boldsymbol{\theta}$. We then update each $\theta_j$ by $\theta_j \leftarrow \text{rnd}_{\text{fp32}}(\theta_j - \eta \cdot \hat{d\theta}_j)$ given a learning rate $\eta > 0$, and proceed to the next iteration until the training ends. Here we use fp32 to represent $\theta_j$ by following the convention in low-precision floating-point training: a "master copy" of weights (i.e., $\theta_j$) is stored separately from the weight values (i.e., $\hat{\theta}_j$) used in a gradient computation, and is usually represented by fp32 (Cambier et al., 2020; Kalamkar et al., 2019; Micikevicius et al., 2018). The memory overhead of this master copy is very small compared to the memory required to store other tensors (e.g., activation tensors $v_i$) (Micikevicius et al., 2018).

**Low-precision floating-point training.** In low-precision training, we use a precision assignment $\pi$ where some tensors have a smaller bitwidth than fp32. Particularly well-studied are $\pi$ that use two predetermined floating-point bitwidths (which are different) and optionally vary the rest of the format from tensor to tensor. We call $\mathcal{C} : \text{TS} \times \{\text{lo}, \text{hi}\} \to \text{FP}$ a *precision-candidate assignment* if $\mathcal{C}(t, \text{lo})$ has the same bitwidth for all $t \in \text{TS}$, the same holds for hi, and the bitwidth for lo is smaller than that for hi. We define $\Pi(\mathcal{C}) \triangleq \{\pi : \text{TS} \to \text{FP} \mid \forall t \in \text{TS}. \, \pi(t) \in \{\mathcal{C}(t, \text{lo}), \mathcal{C}(t, \text{hi})\}\}$ as the set of precision assignments that conform to $\mathcal{C}$.

Among various precision assignments in $\Pi(\mathcal{C})$, two have received the most attention: the uniform assignment $\pi_{\text{unif},\mathcal{C}}$ (Micikevicius et al., 2018) and the operator-based assignment $\pi_{\text{op},\mathcal{C}}$ (Sun et al., 2019). The former assigns low-precision formats to all tensors uniformly[1], and the latter to (most of) the input tensors of GEMM operators (in both forward and backward passes):

$$\begin{aligned}
\pi_{\text{unif},\mathcal{C}}(t) &\triangleq \mathcal{C}(t, \text{lo}) \text{ for all } t \in \text{TS}, \\
\pi_{\text{op},\mathcal{C}}(t) &\triangleq \begin{cases} \mathcal{C}(t, \text{lo}) & \text{if } t \in \{v_i, \theta_i, dv_{i+1}\} \text{ for some } i \\ & \quad \text{and } f_i \text{ is a GEMM operator (but not the first/last one)} \\ \mathcal{C}(t, \text{hi}) & \text{otherwise,} \end{cases}
\end{aligned} \tag{2}$$

where a GEMM operator refers to a general matrix multiplication operator which arises in, e.g., fully-connected or convolutional layers. A particular variant $\pi_{\text{op}',\mathcal{C}}$ of $\pi_{\text{op},\mathcal{C}}$ has received much attention as well (Kalamkar et al., 2019), which assigns low-precision formats to (most of) the input and output tensors of GEMM operators: it is defined as $\pi_{\text{op},\mathcal{C}}$ except that $\{v_i, \theta_i, dv_{i+1}\}$ in Eq. (2) is replaced by $\{v_i, \theta_i, v_{i+1}, dv_i, d\theta_i, dv_{i+1}\}$.

---

[1]For simplicity we define $\pi_{\text{unif},\mathcal{C}}$ without the common exceptions for tensors near $v_1$ and/or $v_{m+1}$.

We note that the precision assignments used in `apex.amp` and `torch.amp` (Nvidia, 2019; PyTorch, 2022) correspond to $\pi_{\mathrm{op},\mathcal{C}}$ and $\pi_{\mathrm{op}',\mathcal{C}}$, respectively. For several choices of $\mathcal{C}$, these assignments have been shown to produce training accuracy similar to that by $\pi_{\mathrm{fp32}}$ on many datasets and models (see §1–2).

### 3.2 Memory-Accuracy Tradeoff Problem

We now introduce the following problem based on §3.1, to address the limitation of existing approaches for low-precision floating-point training discussed in §1:

**Problem 3.1** (Memory-accuracy tradeoff). Given training data $\{(x_i, y_i)\}$, a model and loss network $\mathcal{M}$ and $\mathcal{L}$, a precision-candidate assignment $\mathcal{C}$, and a lower bound $r \in [0, 1]$ on the low-precision ratio, find a precision assignment $\pi \in \Pi(\mathcal{C})$ that maximizes $\mathrm{acc}(\pi)$ subject to $\mathrm{ratio}_{\mathsf{lo}}(\pi) \geq r$. $\qquad\square$

Here $\mathrm{acc}(\pi)$ denotes the accuracy of the model $\mathcal{M}$ when trained with $\pi$ on $\{(x_i, y_i)\}$, and $\mathrm{ratio}_{\mathsf{lo}}(\pi)$ denotes the *low-precision ratio* of $\pi$, i.e., the portion of the tensors represented in low-precision under $\pi$, among all tensors appearing in a gradient computation:

$$\mathrm{ratio}_{\mathsf{lo}}(\pi) \triangleq \frac{\mathrm{size}(\{t \in \mathsf{TS} \mid \pi(t) = \mathcal{C}(t, \mathsf{lo})\})}{\mathrm{size}(\mathsf{TS})} \in [0, 1]$$

where $\mathrm{size}(T) \triangleq \sum_{t \in T} \mathrm{size}(t)$ denotes the total size (i.e., number of elements) of all tensors in $T \subseteq \mathsf{TS}$.[2] For instance, $\mathrm{ratio}_{\mathsf{lo}}(\pi_{\mathsf{hi}}) = 0$ and $\mathrm{ratio}_{\mathsf{lo}}(\pi_{\mathsf{lo}}) = 1$ for the all-high-precision assignment $\pi_{\mathsf{hi}}$ and the all-low-precision assignment $\pi_{\mathsf{lo}}$. The problem asks for a precision assignment that maximizes training accuracy under a memory constraint, which is expressed as a fraction of the memory required to train the model using $\pi_{\mathsf{hi}}$.

### 3.3 NP-Hardness of the Problem

We prove that the memory-accuracy tradeoff problem from §3.2 is NP-hard by showing that there is a polynomial-time reduction from the knapsack problem to this problem:

**Theorem 3.2.** *Problem 3.1 is NP-hard.*

*Proof sketch.* Recall the knapsack problem: given $n$ items with weights $w_i \in \mathbb{N}$ and profits $p_i \in \mathbb{N}$ ($i \in [n]$), find a subset of the items that maximizes the total profit while its total weight does not exceed a given threshold $W \in \mathbb{N}$.

Given an instance $(w, p, W)$ of the knapsack problem, we construct an instance of Problem 3.1 such that we get the following (informal) correspondence between the two: $w_i$ corresponds to the size of the parameter tensor $\theta_i$; $p_i$ to the $i$-th component of the input data; $W$ to the lower bound $r$ on the low-precision ratio (in an inverse way); and selecting the $i$-th item corresponds to assigning a high-precision format to the tensor $\theta_i$ (and related tensors), which roughly decreases the low-precision ratio by $w_i$ while increasing the accuracy of the model (after training) by $p_i$. Based on this informal correspondence, we formally prove that an optimal solution to the above instance of Problem 3.1 can be converted in linear time to an optimal solution to the given knapsack problem $(w, p, W)$. That is, we have a linear-time reduction from the knapsack problem (which is NP-hard (Karp, 1972)) to Problem 3.1 which is therefore NP-hard. For a detailed proof, see Appendix A. $\quad\square$

Intuitively, the proof relies on two aspects of Problem 3.1: the size of the search space (i.e., $|\Pi(\mathcal{C})|$) is exponential in the size of the problem (especially $|\mathsf{TS}|$), and some values representable in a high-precision format underflow to 0 in a lower-precision format. Note that underflows are relevant in low-precision training: they frequently arise in practice, degrading the results of training (Micikevicius et al., 2018). The NP-hardness result indicates that it is unlikely any polynomial-time algorithm solves the problem exactly.

## 4 Algorithm

In this section, we propose a novel (heuristic) algorithm for the memory-accuracy tradeoff problem (§4.1), and a new technique to handle overflows arising in training (§4.2). We point out that the former algorithm

---

[2]As explained in §1, the low-precision ratio is a proxy for the reduction in memory as well as training time (because the low-precision ratio increases as the model aggregate decreases).

finds an initial precision assignment *before* training starts, whereas the latter technique updates the current precision assignment *while* training proceeds.

## 4.1 Precision Demotion for Saving Memory

Consider an input to the memory-accuracy trade-off problem (Problem 3.1): a model and loss network $\mathcal{M} = (f_1, \ldots, f_n)$ and $\mathcal{L} = (f_{n+1}, \ldots, f_m)$, a precision-candidate assignment $\mathcal{C}$, and a lower bound $r$ on the low-precision ratio. Given the input, our algorithm finds a precision assignment $\pi$ in two steps (Algorithm 1).

**Tensor grouping.** We first *group* tensors in TS such that each group consists of all the tensors between two "adjacent" GEMM operators (see below for details). This grouping reduces the search space over precision assignments, from all of $\Pi(\mathcal{C})$ to a subset in which the same precision is assigned to the tensors in the same group. This specific grouping strategy is based on two observations: a majority of floating-point operations are carried out in GEMM operators, and it is standard (e.g., in PyTorch) to use the same precision for a forward tensor and its corresponding backward tensor.

Formally, we group tensors as follows. Let $f_k$ and $f_{k'}$ $(k < k')$ be GEMM operators that are "adjacent", i.e., there is no GEMM operator in $\{f_{k+1}, \ldots, f_{k'-1}\}$. For each such $(f_k, f_{k'})$, we create a group $\{v_i, dv_i, \theta_j, d\theta_j \mid i \in (k, k'] \cap [m+1], j \in (k, k'] \cap [n]\}$. After that, we create two more groups for the remaining tensors: one for the tensors near $v_1$ and the other for tensors near $v_{m+1}$. As a result, we obtain a set of disjoint groups of tensors $\{T_1, T_2, \ldots\} \subseteq 2^{\mathsf{TS}}$.

**Precision demotion.** Given the groups of tensors, $T_1, T_2, \ldots$, we construct a precision assignment $\pi$ as follows: initialize $\pi$ to the all-high-precision assignment and update $\pi$ by *demoting* the precision of all tensors in a group to low precision, one group at a time, until the low-precision ratio of $\pi$ becomes greater than $r$. We demote the precision of groups in *decreasing* order of their sizes (i.e., the total number of elements in tensors); that is, the precision of a larger size group is demoted earlier. Formally, let $\{T_1', T_2', \ldots\}$ be the reordering of $\{T_1, T_2, \ldots\}$ such that $\mathrm{size}(T_1') \geq \mathrm{size}(T_2') \geq \cdots$. After initializing $\pi$ by $\pi(t) = \mathcal{C}(t, \mathsf{hi})$ for all $t$, we iterate over $i \in \mathbb{N}$ and update $\pi$ to $\pi(t) = \mathcal{C}(t, \mathsf{lo})$ for all $t \in T_i'$, until $\mathrm{ratio}_{\mathsf{lo}}(\pi) \geq r$ is first satisfied. The resulting $\pi$ is the output of our algorithm.

The intuition behind using group size as the priority order for precision demotion is based on the fact that it is actually *optimal* in a very simplified setting. Suppose that an input $x$ to the model $\mathcal{M}$ stores a quantity of information $I$ and the forward computation of $\mathcal{M}$ is nothing but a process of extracting the information in the input into a small number of values, i.e., the tensor $v_{n+1}$. Assume that passing through each group $O_i = \{f_{k+1}, \ldots, f_{k'}\}$ of operators (corresponding to the group $T_i$ of tensors) reduces the amount of information by a factor $\alpha_i \in (0, 1)$, and using low precision on the group $T_i$ further reduces the amount of information by a constant factor $\beta \in (0, 1)$ for all $i$. Then, the amount of information left in $v_{n+1}$ becomes $I \times (\alpha_1 \alpha_2 \cdots) \times \beta^l$, where $l$ is the number of groups in low precision. In this simplified setting, maximizing the amount of information in $v_{n+1}$ is equivalent to minimizing the number of groups in low precision, which is achieved precisely by demoting the largest groups first (when there is a constraint on the low-precision ratio). We show empirically (§5.4) that using the decreasing size order in precision demotion indeed produces better precision assignments than using other orders.

## 4.2 Precision Promotion for Handling Overflows

While our algorithm in §4.1 exerts a constraint on memory usage, it places no explicit constraint on training accuracy, and so not surprisingly for some models and datasets the resulting precision assignment causes training to *diverge*—accuracy decreases significantly and remains low after some point. We observe that when training begins to diverge (and a bit before that), many overflows occur in the rounding function of some tensors $\hat{v}_i$, i.e., an input tensor to the function $\mathrm{rnd}_{\pi(v_i)}(\cdot)$ in Eq. (1) contains many elements whose magnitude is larger than the maximum representable number of the format $\pi(v_i)$ (Figure 6(a-b); §5.4). This rapid increase in overflows in individual tensors is a signal that training may diverge.

**Precision promotion.** Based on this observation, after each gradient computation we update the current precision assignment $\pi$ by *promoting* to high precision (i.e., $\mathcal{C}(t, \mathsf{hi})$) any forward tensor $t$ whose *overflow ratio* is greater than a given threshold $\Theta \in (0, 1)$; this updated precision assignment is used in the next gradient

**Algorithm 1:** Computing $\pi$ with precision demotion

**Input:** $(f_1, \ldots, f_n)$, $(f_{n+1}, \ldots, f_m)$, $\mathcal{C}$, $r$
/* Tensor grouping */
$k = 1; T_k = \emptyset$
**for** $i = 1$ **to** $m$ **do**
    $T_k = T_k \cup \{v_i, dv_i\}$
    **if** $k \leq n$ **then** $\{ T_k = T_k \cup \{\theta_i, d\theta_i\} \}$
    **if** $f_i$ is GEMM **then** $\{ k = k + 1; T_k = \emptyset \}$
**end**
/* Precision demotion */
$(T'_1, \ldots, T'_k) = \text{sort}(T_1, \ldots, T_k)$ by decreasing size
$\pi(t) = \mathcal{C}(t, \mathsf{hi})$ for all $t \in \mathsf{TS}$
**for** $j = 1$ **to** $k$ **do**
    **if** $\text{ratio}_{\mathsf{lo}}(\pi) \geq r$ **then** $\{$ **break** $\}$
    $\pi(t) = \mathcal{C}(t, \mathsf{lo})$ for all $t \in T'_j$
**end**
**return** $\pi$

**Algorithm 2:** Training with precision promotion

**Input:** $\pi$, $\Theta$, $\mathcal{C}$
/* Training loop */
Initialize weights $\boldsymbol{\theta}$
**while** training not finished **do**
    Compute the current gradient $\mathbf{d\theta}$ using $\pi$
    Update $\boldsymbol{\theta}$ using $\mathbf{d\theta}$
    /* Precision promotion */
    **for** forward $t \in \mathsf{TS}$ **do**
        **if** overflow_ratio$(t) > \Theta$ **then**
            $\pi(t) = \mathcal{C}(t, \mathsf{hi})$
        **end**
    **end**
**end**
**return** $\boldsymbol{\theta}$

computation (Algorithm 2). Here the overflow ratio of $t \in \mathsf{TS}$ denotes the number of overflows arising in the rounding function of $\hat{t}$ in Eq. (1), divided by the number of elements in $\hat{t}$. We show empirically (§5.4) that training always converges using this technique and the additional memory cost of promotion is small (in our experiments, $< 3\%$ of the maximum model aggregate[3]). For the experiments, we use $\Theta = 0.01$; in fact we found that a wide range of values for $\Theta$ (0.1, 0.01, and 0.001) all work well. Note that this technique is not specific to our algorithm and can also be applied to other precision assignment methods.

We apply precision promotion only to forward tensors for two reasons. First, *dynamic loss scaling* (Micikevicius et al., 2018; Nvidia, 2019; PyTorch, 2022; Sun et al., 2019) already handles overflows in backward tensors, but not in forward tensors: loss scaling multiplies the backward loss tensor $dv_{m+1}$ by a constant before performing backward computation, to scale up all backward tensors; the dynamic version adjusts the constant during training in a way that avoids overflows in backward tensors. Note that dynamic loss scaling does not affect forward tensors at all. Second, we cannot use a similar idea to handle overflows in forward tensors, because forward tensors are not linear in the input tensor $v_1$ whereas backward tensors are linear in the backward loss tensor $dv_{m+1}$ (by the linearity of differentiation).

Precision promotion incurs little if any computational overhead: checking whether a single rounding operation overflows is cheap, and we only apply rounding functions to the output tensor of an arithmetic-intensive operator (e.g., convolution and batch normalization), amortizing the cost of the overflow checks over a large number of other operations.

## 5 Experiments

In this section, we evaluate our precision assignment technique (developed in §4) on standard training tasks to answer three research questions:

- Does our technique explore the tradeoff between memory and accuracy and achieve a better tradeoff than existing (fixed) precision assignments (§5.3)?
- Are the two main components of our technique, precision demotion/promotion of larger/overflowing tensors, important for good performance (§5.4)?
- How can we choose the parameter $r$ in our technique (i.e., a lower bound on the low-precision ratio) (§5.5)?

---

[3]For each training where our methods (presented in §4) are used, we measure the model aggregate when the training starts and when it ends. We observe that the difference between the two values (averaged over four different runs) is at most 3% of the maximum model aggregate (i.e., the model aggregate when all tensors are in high precision).

### 5.1 Implementation

We have implemented our precision assignment technique using PyTorch (Paszke et al., 2019). Given a model and loss network, and a dataset, our implementation takes as parameters a precision-candidate assignment $\mathcal{C}$ and a lower bound $r$ on the low-precision ratio; it then automatically assigns precisions to tensors (appearing in training) according to our technique and uses those assigned precisions in gradient computations. To make these procedures automatic, our implementation works as follows:

- For each primitive operator in PyTorch (e.g., `torch.nn.Conv2d`), our implementation provides its wrapped version (e.g., `ext3.nn.Conv2d`) which records auxiliary information for our technique (e.g., floating-point format of input/output tensors) and applies proper rounding functions in forward/backward computations based on the auxiliary information. Models should now use the wrapped classes instead of the original ones.
- Our implementation first constructs a computation graph (of a given model and loss network) dynamically by running a forward computation on a minibatch of input data. The computation graph and other information (e.g., each tensor's size) are recorded in the wrapped classes.
- Using the auxiliary information just recorded, our implementation then constructs an initial precision assignment according to §4.1, and starts training with this assignment. During the training, our implementation uses the current precision in gradient computations, and updates it after each gradient computation according to §4.2. We record the precision assignment also in the wrapped classes to automatically apply proper rounding functions in gradient computations.

We simulate low-precision formats used in the experiments (e.g., 8-bit floats) and their operations, with 32-bit floats and 32-bit operations followed by rounding functions as described in Eq. (1); simulating low-precision formats is the standard methodology set by prior works on low-precision training (Cambier et al., 2020; Fox et al., 2021; Kalamkar et al., 2019; Micikevicius et al., 2022) and we simply follow this.[4] We implement the rounding functions based on the QPyTorch library (Zhang et al., 2019), but a few extensions are required, e.g., to support exponent bias and signal overflows for dynamic loss scaling. We automatically apply these rounding functions after each primitive operator, by using PyTorch's hook feature (e.g., `nn.Module.register_*hook`).

### 5.2 Experiment Setups

**Datasets and models.** As benchmarks for our experiments, we use the image classification task and three datasets for the task: CIFAR-10 and CIFAR-100 (Krizhevsky, 2009), and ImageNet (Russakovsky et al., 2015); these task and datasets have been widely used in recent works on low-precision training as a standard choice (Chmiel et al., 2021; Rajagopal et al., 2020; Sakr & Shanbhag, 2019; Wang et al., 2018) and we simply follow this. For the task and datasets, we use four well-known models: SqueezeNet (Iandola et al., 2016), ShuffleNet-v2 (Ma et al., 2018), MobileNet-v2 (Sandler et al., 2018), and ResNet-18 (He et al., 2016); they are chosen since models with relatively few weights, such as these, are generally known to be more difficult to train with low precision than those with more weights (Sun et al., 2019). We considered other tasks (e.g., language modeling) and related models (e.g., RNN/transformer-based models) but did not include them in our experiments because substantial additional implementation effort orthogonal to our main contributions would be required: these models use some PyTorch operators that do not support per-tensor precision assignments,[5] so applying our technique to these models requires significant modifications to PyTorch internals.

**Precision-candidate and precision assignments.** For the experiments, we use the precision-candidate assignment $\mathcal{C}$ studied in Sun et al. (2019), which uses 16-bit (and 8-bit) floats for high (and low) precision; in particular, $\mathcal{C}(t, \mathsf{hi}) = \mathsf{fp}(6, 9, 0)$ for all (forward/backward) tensors $t$, and $\mathcal{C}(t, \mathsf{lo}) = \mathsf{fp}(4, 3, 4)$ for all forward tensors $t$ and $\mathsf{fp}(5, 2, 0)$ otherwise. We choose this particular $\mathcal{C}$ since it uses sub-32-bit floating-point formats for both low and high precision and the precision assignment $\pi_{\mathrm{op}, \mathcal{C}}$ was shown to achieve accuracy comparable to 32-bit training (Sun et al., 2019). The three floating-point formats used in $\mathcal{C}$ have subnormals but no

---

[4]The 8-bit formats used in our experiments (see §5.2) began to be supported natively in hardware very recently (by NVIDIA H100 GPU). But access to such hardware is still very limited (e.g., no major cloud services including AWS, Azure, and Google Cloud provide it), and these formats are not yet supported natively in software (e.g., PyTorch, TensorFlow, and cuDNN). Due to such lack of a hardware and software ecosystem natively supporting these formats, we chose to simulate them as in prior works.

[5]For instance, the PyTorch functions `nn.RNN` and `nn.MultiheadAttention` do not allow to change the precision of intermediate tensors (e.g., input/output tensors of each GEMM operator used in the functions) to user-defined formats (e.g., $\mathsf{fp}(4, 3, 4)$).

infinities and NaNs, which are rounded to the largest or smallest representable numbers. Since our technique is parameterized by a precision-candidate assignment, it is easily applied to other assignments as well.

We evaluate our technique by varying its parameter $r$ (i.e., a lower bound on low-precision ratio) over deciles $r \in \{0, 0.1, 0.2, \ldots, 1\}$. We write $\pi_{\text{ours},r}$ to denote the precision assignment chosen by our technique (described in §4) for a given $r$; e.g., $\pi_{\text{ours},0}$ is the all-high-precision assignment, and $\pi_{\text{ours},1}$ is the all-low-precision assignment equipped with our precision promotion technique (§4.2). Following Sun et al. (2019), all precision assignments (including $\pi_{\text{ours},r}$) in our experiments use high precision (i.e., 16 bits) for all backward weight tensors (i.e., $\hat{d\theta}_j$). For each precision assignment $\pi$, its low-precision ratio can change during training due to our precision promotion technique (when applied), so we compute the average of the ratio over all epochs and report this value as the low-precision ratio of $\pi$.

**Other setups and compute time.** All experiments were performed on NVIDIA V100 GPUs; total compute time for all experiments was 1,081 GPU days. We train all models in a standard way: we apply dynamic loss scaling (a standard technique used in low-precision floating-point training; see §4.2 for details) except for 32-bit training, and use standard settings (e.g., learning rate); see Appendix B for details. Due to random variations in training, we perform four runs of training for each configuration and report the average and the range of measured quantities.

## 5.3 Comparison with Existing Precision Assignments

To compare our technique with existing precision assignments for floating-point training, we train each model with the following precision assignments: all-32-bit $\pi_{\text{fp32}}$, uniform $\pi_{\text{unif}}$ (Micikevicius et al., 2018), operator-based $\pi_{\text{op}}$ (Nvidia, 2019; Sun et al., 2019), its variant $\pi_{\text{op}'}$ (Kalamkar et al., 2019; PyTorch, 2022), and ours $\pi_{\text{ours},r}$ (see §3.1 and §5.2 for their definitions). We choose $\pi_{\text{unif}}$, $\pi_{\text{op}}$, and $\pi_{\text{op}'}$ as baselines because existing precision assignments for floating-point training fall into one of the three assignments (or their variants) (see §1–2).

We train the four models mentioned in §5.2 on CIFAR-10 and CIFAR-100, and ShuffleNet-v2 on ImageNet. We also train smaller variants of the four models (which are more difficult to train with low precision) on CIFAR-100. We obtain these variant models by following Sun et al. (2019), i.e., by applying a well-known approach for model reduction that uses a parameter called the *width multiplier* (Howard et al., 2017): each variant model reduces the number of channels in most tensors by a width multiplier; we use three values $\{0.5, 0.25, 0.1\}$ for the width multiplier. We train just one model on ImageNet due to the large amount of computation involved: for each model, 44 training runs (11 choices for $r$ and 4 runs for each choice) are required for $\pi_{\text{ours},r}$ and each run on ImageNet takes nearly a half day with 16 GPUs. We use ShuffleNet-v2 for ImageNet since the model shows interesting memory-accuracy tradeoffs when trained on the (smaller) CIFAR datasets.

**ImageNet.** Figure 3 presents training results of ShuffleNet-v2 on ImageNet: its left graph plots the average training trajectory for each precision assignment, and its right graph shows how each precision assignment trades off between memory and accuracy, where memory is represented (inversely) by the low-precision ratio of the assignment (which is averaged over all epochs; see §5.2) and accuracy is the best test accuracy of the model during training. Each point in the right graph shows the average accuracy of four runs of training, while the shaded area shows the variation in accuracy among those four training runs.

Figure 3 shows three points. First, as the parameter $r$ increases, the average accuracy drop of $\pi_{\text{ours},r}$ from $\pi_{\text{fp32}}$ increases (up to 5%). In contrast, $\pi_{\text{unif}}$ and $\pi_{\text{op}'}$ have a much larger average accuracy drop (more than 30%), as some training runs diverge when $\pi_{\text{unif}}$ and $\pi_{\text{op}'}$ are used. Second, the tradeoff given by $\pi_{\text{ours},r}$ is better (i.e., closer to Pareto-optimal) than by $\pi_{\text{op}}$: $\pi_{\text{ours},r}$ for $r \in \{0.3, 0.4\}$ has both higher accuracy and larger low-precision ratio (i.e., memory reduction) than $\pi_{\text{op}}$. In particular, $\pi_{\text{ours},0.4}$ has $1.6\times$ the memory reduction of $\pi_{\text{op}}$. Third, $\pi_{\text{ours},r}$ provides options that $\pi_{\text{op}}$ cannot (which has an accuracy drop of $>1\%$). If we want accuracy closer to $\pi_{\text{fp32}}$, say within 0.5%, we can use $\pi_{\text{ours},0.2}$ with 2.6% more memory than $\pi_{\text{op}}$. If we can tolerate a larger accuracy loss, say $\approx 3\%$, then we can use $\pi_{\text{ours},0.7}$ with $2.9\times$ the memory reduction of $\pi_{\text{op}}$.

**CIFAR-10/100.** Figure 4 presents the memory-accuracy tradeoffs of precision assignments for the four models on CIFAR-10 and CIFAR-100, and their smaller variants (with width multiplier 0.25) on CIFAR-100. The results for other smaller variants are similar and included in Figure 10 (see Appendix C.1).

The conclusions from Figure 3 hold for Figure 4: $\pi_{\text{ours},r}$ provides a range of options by varying $r$ and exhibits a better tradeoff than $\pi_{\text{unif}}$, $\pi_{\text{op}}$, and $\pi_{\text{op}'}$ in almost all cases. We give a detailed comparison as follows. First, in half of all 12 plots, $\pi_{\text{unif}}$ shows a similar tradeoff to $\pi_{\text{ours},1}$. But in the remaining half, $\pi_{\text{unif}}$ has an accuracy drop much larger than all other precision assignments including $\pi_{\text{ours},r}$, since using $\pi_{\text{unif}}$ often makes training diverge while using, e.g., $\pi_{\text{ours},1}$ does not do so. Second, in all but two plots, $\pi_{\text{ours},r}$ shows a strictly better tradeoff than $\pi_{\text{op}}$: $\pi_{\text{ours},r}$ has noticeably larger ($> 2\times$) memory reduction than $\pi_{\text{op}}$ while maintaining similar accuracy. Even in the two plots, $\pi_{\text{ours},r}$ has a tradeoff very close to $\pi_{\text{op}}$. Note that in three plots, $\pi_{\text{op}}$ has an accuracy drop of $>1\%$ while $\pi_{\text{ours},r}$ provides several options that have smaller accuracy drops and more memory savings at the same time. Third, $\pi_{\text{ours},r}$ shows a strictly better (or similar) tradeoff than $\pi_{\text{op}'}$ in all but two (or two) plots. Note that $\pi_{\text{op}'}$ has accuracy smaller than $\pi_{\text{op}}$ in all but one plots. Also it has an accuracy drop of $>1\%$ in half of all plots, and sometimes makes training even diverge (in one plot here and three other plots in Figure 10).

**Additional results.** To isolate the effect of our precision promotion technique on the above results (Figures 3, 4 and 10), we compare our precision assignments and the baseline assignments while applying the precision promotion to all of them, and present the results in Appendix C.1 (Figures 13–15). Two observations can be made in these results: for each precision assignment, (i) if training diverged without the precision promotion, applying the precision promotion prevents such divergence and produces much higher accuracy; (ii) otherwise, the accuracy and the low-precision ratio remain similar regardless of using the precision promotion. These observations lead to the same conclusion as in the above: our assignments provide similar or better tradeoff between memory and accuracy than the baseline assignments, even when the latter are equipped with our precision promotion technique. In addition, these observations also indicate that our precision promotion technique can effectively handle divergence in training (see §5.4 for more results on this).

### 5.4 Ablation Study: Precision Demotion and Promotion

**Precision demotion.** To evaluate the decision to use precision demotion in decreasing-size order, we train the four models on CIFAR-100 with $\pi_{\text{ours},r}$, $\pi_{\text{ours[inc]},r}$ (which demotes tensor groups in increasing-size order) and $\pi_{\text{ours[rand]},r}$ (which demotes tensor groups in random order). For $\pi_{\text{ours[rand]}}$, three different random orders are used in each case. The results, presented in Figure 5 (and Appendix C.2), show that the order of precision demotion has a significant impact on the resulting memory-accuracy tradeoff, and that decreasing order provides the best results in nearly all cases. Increasing order consistently shows the worst results, suggesting our intuition (given in §4.1) for choosing decreasing order has some basis in reality.

**Precision promotion.** To understand whether precision promotion of overflowing tensors is important to our technique, we train ShuffleNet-v2 on ImageNet using $\pi_{\text{ours[no-promo]},r}$ which does not promote tensors. The results, presented in Figure 6(a), show that several training trajectories diverge in early epochs and fail to recover afterwards. Figure 6(b) plots the top-5 tensor overflow ratios for the highlighted trajectory in Figure 6(a). The overflow ratios first spike about when divergence occurs around epoch 11. A closer look shows that the spike in overflow ratio occurs shortly before divergence, and starts first in a few tensors and then propagates to others. These observations indicate that an excessive number of overflows in a few tensors are the cause of the training divergence.

Finally, Figure 6(c-d) shows that precision promotion is effective at preventing the divergence of training while sacrificing only a small amount of memory reduction. The figure shows ShuffleNet-v2 on ImageNet trained using our technique with and without precision promotion. Figure 6(c) shows that without precision promotion large accuracy drops occur due to divergence, whereas with precision promotion training converges. Figure 6(d) shows that the total size of tensors promoted to high precision is small for all $r$ values. See Appendix C.2 for similar results for CIFAR-10.

### 5.5 Choosing the Value of $r$

The time and space savings of our method are most significant when a model is regularly retrained, which commonly occurs when new data is periodically incorporated into an existing model. Assuming that new data has a similar distribution to existing data, we can choose a single $r$ (a parameter in our method) by conducting one set of experiments where we train with $\pi_{\text{fp32}}$ and $\pi_{\text{ours},r}$ for different $r$ and then choose the $r$ value that maximizes model aggregate savings while still having an acceptable drop in accuracy.

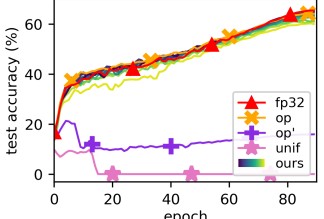 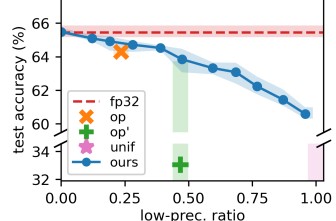

Figure 3: Results of training ShuffleNet-v2 on ImageNet with $\pi_{\text{fp32}}$, $\pi_{\text{unif}}$ (Micikevicius et al., 2018), $\pi_{\text{op}}$ (Sun et al., 2019), $\pi_{\text{op}'}$ (Kalamkar et al., 2019), and $\pi_{\text{ours},r}$. Left: Each line shows the average training trajectory for each precision assignment; $\pi_{\text{ours},r}$ is colored from navy to yellow (darker for smaller $r$). A zoomed-in version of this plot can be found in Appendix C.1. Right: Each point shows the memory-accuracy tradeoff of each precision assignment; a red-dashed line shows the accuracy of $\pi_{\text{fp32}}$; and shaded areas show the variation among four training runs. In the right figure, top-right points are better than bottom-left ones. Observe that there are ●s above and to the right of ✖ and ✚, respectively. ★ is missing as its y-value is too small.

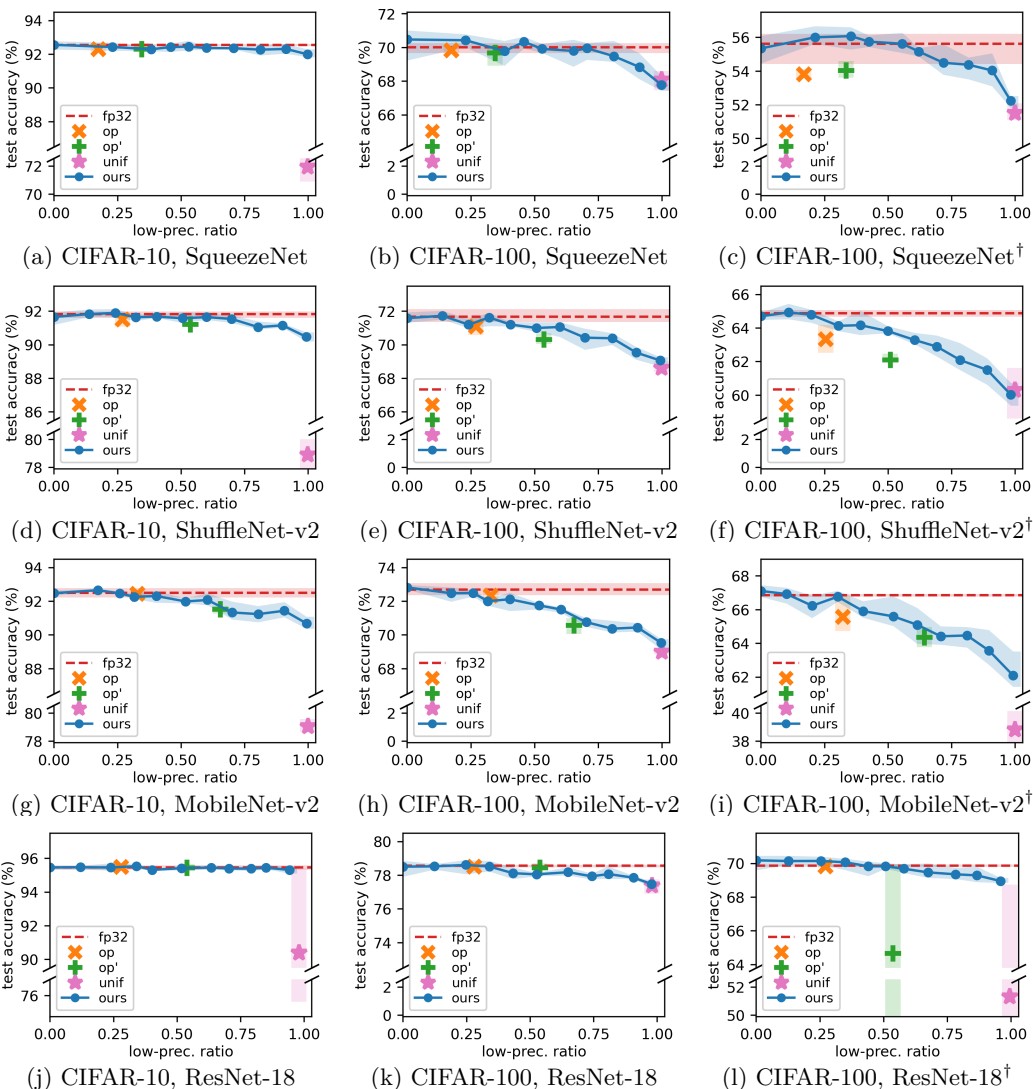

Figure 4: Memory-accuracy tradeoffs of $\pi_{\text{unif}}$ (Micikevicius et al., 2018), $\pi_{\text{op}}$ (Sun et al., 2019), $\pi_{\text{op}'}$ (Kalamkar et al., 2019), and $\pi_{\text{ours},r}$ for four models and their smaller variants on CIFAR-10 and CIFAR-100. The variant models have width multiplier 0.25 and are marked by †. Top-right points are better than bottom-left ones. In all but three plots, there are ●s above and to the right of ✖ and ✚, respectively; even in the three plots (g,h,k), ●s have almost the same tradeoffs to ✖ and ✚. In half of all plots, ★ has much smaller y-values than other points. The training trajectories for the above plots and the results of other smaller models are in Appendix C.1.

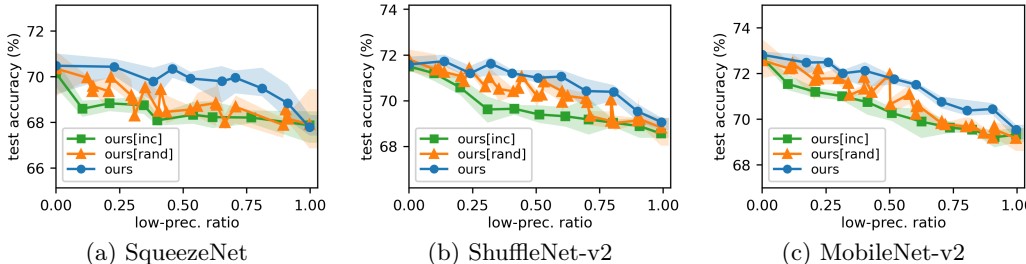

(a) SqueezeNet     (b) ShuffleNet-v2     (c) MobileNet-v2

Figure 5: Memory-accuracy tradeoffs of $\pi_{\mathrm{ours},r}$, $\pi_{\mathrm{ours[inc]},r}$, and $\pi_{\mathrm{ours[rand]},r}$ for three models on CIFAR-100. Observe that ●s are above and to the right of other points in nearly all cases. The results of ResNet-18 are in Appendix C.2.

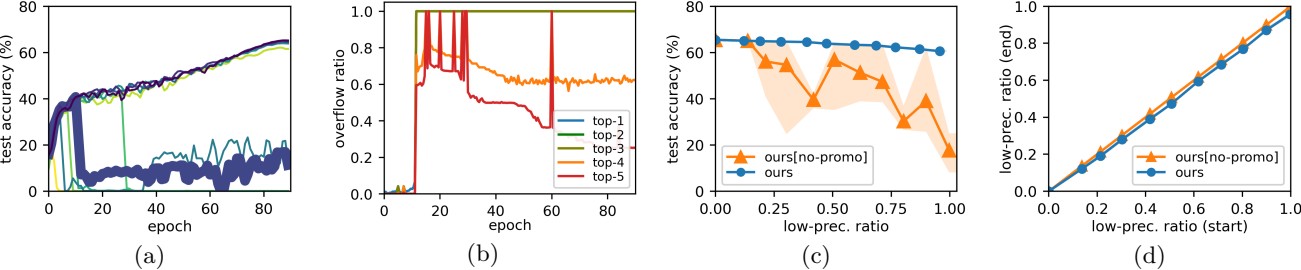

(a)     (b)     (c)     (d)

Figure 6: Training ShuffleNet-v2 on ImageNet with $\pi_{\mathrm{ours},r}$ and $\pi_{\mathrm{ours[no-promo]},r}$. (a) Training trajectories of $\pi_{\mathrm{ours[no-promo]},r}$ for different $r$; colors denote $r$ values (darker for smaller $r$). (b) Top-5 overflow ratios of tensors at each epoch, for the highlighted trajectory in (a); the largest ratio is blue and the fifth largest red. (c) Memory-accuracy tradeoffs of $\pi_{\mathrm{ours},r}$ and $\pi_{\mathrm{ours[no-promo]},r}$. (d) Low-precision ratio when training ends vs. when training starts, for $\pi_{\mathrm{ours},r}$ and $\pi_{\mathrm{ours[no-promo]},r}$. The results on CIFAR-10 are in Appendix C.2.

To simulate this scenario, we create five datasets ImageNet-200-$i$ ($i \in [5]$) as follows, so that each of them contains different but similar data: randomly select 1/5 of the classes in ImageNet (which has 1000 classes in total), and split the training data of each class evenly into five new datasets.

For each ImageNet-200-$i$, we train ShuffleNet-v2 with $\pi_{\mathrm{fp32}}$ and $\pi_{\mathrm{ours},r}$ and present the results in Figure 7. Based on the tradeoff results of $\pi_{\mathrm{ours},r}$, we can choose $r = 0.4$ if we desire an average of $< 1\%$ accuracy drop from $\pi_{\mathrm{fp32}}$, and we can choose $r = 0.9$ if an average $\approx 3\%$ accuracy drop is tolerable. We make two more observations: the tradeoff result of $\pi_{\mathrm{ours},r}$ is similar across all five datasets even though each dataset is different, and for each $r$ the variance in the accuracy of $\pi_{\mathrm{ours},r}$ from different datasets and runs of training is similar to that of $\pi_{\mathrm{fp32}}$. Thus we expect that on a new but similar dataset, $\pi_{\mathrm{ours},r}$ would have an accuracy drop similar to Figure 7 with acceptable variance.

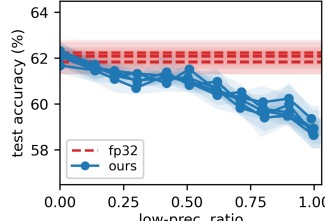

Figure 7: Memory-accuracy tradeoffs of $\pi_{\mathrm{ours},r}$ for ShuffleNet-v2 on ImageNet-200-$i$ ($i \in [5]$).

## 6 Limitations and Future Work

Our work has the same limitation present in prior works on low-precision floating-point training: low-precision floats and operations are simulated in software (instead of being handled natively in hardware) and so the potential speedup of our method is not directly measured, though we do expect speedups to be proportional to the reduction in the model aggregate. We leave it as future work to perform such experiments on very recent or future hardware (e.g., NVIDIA H100 GPU) that natively supports more low-precision formats. Another direction for future work is to integrate our method into systems for automatically optimizing deep learning computations (e.g., Jia et al. (2019); Unger et al. (2022)) to accelerate training.

## Acknowledgments

We thank Thiago S. F. X. Teixeira for helping us run the experiments on cluster machines, and Colin Unger for providing helpful comments on an earlier version of this paper. This work was supported in part by the

Advanced Simulation and Computing (ASC) program of the US Department of Energy's National Nuclear Security Administration (NNSA) via the PSAAP-III Center at Stanford, Grant No. DE-NA0002373.

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

# A    Problem: Deferred Proof

**Theorem 3.2.** *Problem 3.1 is NP-hard.*

*Proof.* We prove the NP-hardness of Problem 3.1 (the memory-accuracy tradeoff problem) by reducing the knapsack problem (which is NP-hard) to the tradeoff problem. More precisely, we prove that the knapsack problem can be solved in polynomial time if we assume an oracle for the tradeoff problem.

Recall the knapsack problem: given $n$ items with weights $w_i \in \mathbb{N}$ and profits $p_i \in \mathbb{N}$ ($i \in [n]$), and given a threshold $W \in \mathbb{N}$, decide which items to choose such that the total profit of the chosen items is maximized while their total weight does not exceed $W$. That is, find $\alpha \in \{0,1\}^n$ that maximizes $\sum_{i \in [n]} \alpha_i p_i$ subject to $\sum_{i \in [n]} \alpha_i w_i \leq W$. This problem is well-known to be NP-hard (Karp, 1972).

Given an instance of the knapsack problem $(w, p, W)$, we construct an instance of the tradeoff problem as follows.

- **Notations.** The following construct uses a constant $k \in \mathbb{N}$ and floating-point formats $\mathsf{fp_{hi}}, \mathsf{fp_{lo}} \in \mathsf{FP}$ (one for high precision and the other for low precision). Below we will specify the conditions they should satisfy, and show that some $k$, $\mathsf{fp_{hi}}$, and $\mathsf{fp_{lo}}$ indeed satisfy the conditions. We write $\mathrm{rnd_{hi}}(\cdot)$ and $\mathrm{rnd_{lo}}(\cdot)$ as shorthand for $\mathrm{rnd_{fp_{hi}}}(\cdot)$ and $\mathrm{rnd_{fp_{lo}}}(\cdot)$.
- **Training setups.** We consider a very simple setting for training: the gradient descent algorithm with a learning rate $\eta = 2^{-l}$ ($l \in \mathbb{N}$) is applied for just one epoch; all parameters are initialized to 0 and their master copies are represented in $\mathsf{fp_{hi}}$; and the negative loss of a model on training data (i.e., $-L(f_\theta(x), y)$ using notations to be described below) is used as the accuracy of the model. Here $l \in \mathbb{N}$ can be any natural number.
- **Model and loss networks.** A model network $\mathcal{M}$ and a loss network $\mathcal{L}$ are given as Figure 8, where $\mathcal{M}$ has $n$ parameter tensors $\theta_i \in \mathbb{R}^{w_i}$ of size $w_i$ ($i \in [n]$). For an input-output pair $(x, y) \in \mathbb{R}^n \times \mathbb{R}$, $\mathcal{M}$ and $\mathcal{L}$ compute a predicted output $f_\theta(x) \in \mathbb{R}$ and a loss $L(f_\theta(x), y) \in \mathbb{R}$ as follows (assuming that no rounding functions are applied):

$$f_\theta(x) \triangleq \sum_{i \in [n]} \sum_{j \in [w_i]} \theta_{i,j} x_i, \qquad\qquad L(f_\theta(x), y) \triangleq 2^{-k} |f_\theta(x) - y|.$$

  Roughly speaking, $\mathcal{M}$ is (a variant of) a linear classifier and $\mathcal{L}$ is a $\ell_1$-loss (scaled by $2^{-k}$).
- **Training data.** Training data consists of a single input-output pair $(x, y) \in \mathbb{R}^n \times \mathbb{R}$ that satisfies the following:

$$x_i = \mathrm{rnd_{lo}}(\sqrt{p_i/w_i}), \qquad\qquad y < -2^{-(k+l)} \sum_{i \in [n]} w_i x_i^2$$

  for all $i \in [n]$. Here $y$ can take any value as long as it satisfies the above inequality. Note that $x_i$ can be different from $\sqrt{p_i/w_i}$ since the latter value may not be representable in $\mathsf{fp_{lo}}$.
- **Precision-candidate assignment.** A precision-candidate assignment $\mathcal{C} : \mathsf{TS} \times \{\mathsf{hi}, \mathsf{lo}\} \to \mathsf{FP}$ is given as:

$$\mathcal{C}(t, \mathsf{hi}) \triangleq \mathsf{fp_{hi}}, \quad \mathcal{C}(t, \mathsf{lo}) \triangleq \mathsf{fp_{lo}} \quad \text{for all } t \in \mathsf{TS}.$$

  That is, for all tensors, $\mathsf{fp_{hi}}$ is used as a high-precision format and $\mathsf{fp_{hi}}$ as a low-precision format. Here $\mathsf{fp_{hi}}$ and $\mathsf{fp_{lo}}$ should satisfy the following:

$$e_{\mathsf{hi}} \geq e_{\mathsf{lo}}, \quad m_{\mathsf{hi}} \geq m_{\mathsf{lo}}, \tag{3}$$

$$|\mathrm{rnd_{lo}}(s) - s| < |s| \cdot err \quad \text{for all } s \in S_1, \tag{4}$$

$$\mathrm{rnd_{lo}}(s) = 0 \qquad\quad \text{for all } s \in S_2, \tag{5}$$

$$\mathrm{rnd_{hi}}(s) = s \qquad\quad \text{for all } s \in S_2 \cup S_3. \tag{6}$$

  Here $e_{\mathsf{hi}}$ and $m_{\mathsf{hi}}$ (and $e_{\mathsf{lo}}$ and $m_{\mathsf{lo}}$) denote the number of exponent bits and mantissa bits of $\mathsf{fp_{hi}}$ (and $\mathsf{fp_{lo}}$), and $err$ and $S_j$ are defined as: $err \triangleq 1/(6n \cdot \max_{i \in [n]} p_i)$, $S_1 \triangleq \{\sqrt{p_i/w_i} \mid i \in [n]\}$, $S_2 \triangleq \{2^{-k}\} \cup \{2^{-k} x_i \mid i \in [n]\}$, and $S_3 \triangleq \{2^{-(k+l)} x_i \mid i \in [n]\}$. Eq. (4) says that the relative error of representing each $s \in S_1$ in $\mathsf{fp_{lo}}$ should be less than $err$. Eq. (5) says that each $s \in S_2$ should underflow to 0 when represented in $\mathsf{fp_{lo}}$. Eq. (6) says that each $s \in S_2 \cup S_3$ should be representable in $\mathsf{fp_{hi}}$.

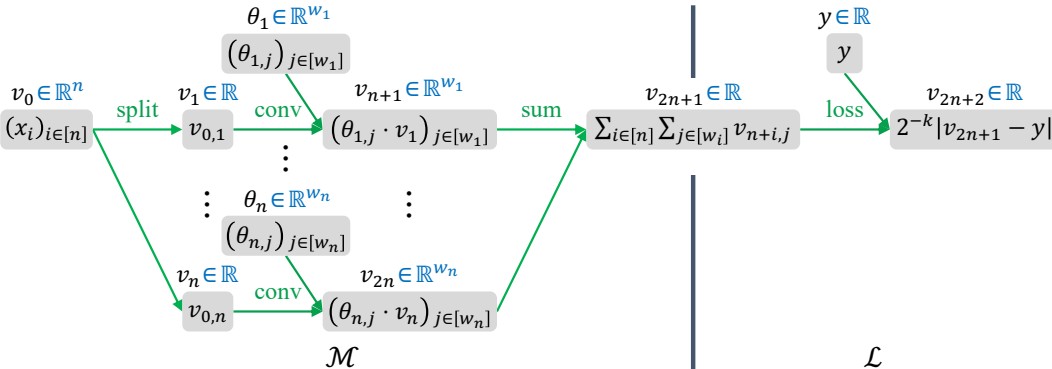

Figure 8: The model network $\mathcal{M}$ and the loss network $\mathcal{L}$ used in the proof of Theorem 3.2.

- **Low-precision ratio.** A lower bound $r \in [0,1]$ on the low-precision ratio is given as:

$$r \triangleq \max\left\{0, 1 - \frac{2W+1}{\mathrm{size}(\mathsf{TS})}\right\} \in [0,1].$$

So $r$ decreases linearly as $W$ increases.

We make three points on the above construction.

- First, each part of the knapsack problem $(w, p, W)$ is used in the following parts of the construction: $w_i$ is used mainly in the size of the parameter tensor $\theta_i$; $p_i$ in the input $x_i$; and $W$ in the lower bound $r$.
- Second, there exist $k \in \mathbb{N}$ and $\mathsf{fp_{hi}}, \mathsf{fp_{lo}} \in \mathsf{FP}$ that satisfy Eqs. (3)–(6). This can be shown as follows: first, by taking sufficiently many exponent and mantissa bits for $\mathsf{fp_{lo}}$, we can make Eq. (4) satisfied; next, by taking a sufficiently large $k$, we can make Eq. (5) satisfied; finally, by taking sufficiently many exponent and mantissa bits for $\mathsf{fp_{hi}}$, we can make Eq. (3) and Eq. (6) satisfied (since $x_i$ is representable in $\mathsf{fp_{lo}}$ and $2^{-(k+l)}$ is a power of two).
- Third, some well-known models (e.g., ShuffleNet-v2) have a similar structure to $\mathcal{M}$ in that they apply the following operations as a subroutine: split a tensor into multiple tensors, apply some operators to each split tensor, and combine the resulting tensors into a single tensor.

We now prove that the knapsack problem $(w, p, W)$ can be solved in polynomial time, if an oracle to the above tradeoff problem is given. Suppose that $\pi \in \Pi(\mathcal{C})$ is an optimal solution to the above tradeoff problem (given by the oracle). Define an item selection $\alpha \in \{0,1\}^n$ for the knapsack problem as:

$$\alpha_i \triangleq \begin{cases} 1 & \text{if } \pi(d\theta_i) = \pi(dv_{n+i}) = \pi(dv_{2n+1}) = \mathsf{fp_{hi}} \\ 0 & \text{otherwise} \end{cases}$$

for each $i \in [n]$. Note that $\alpha$ can be constructed from $\pi$ in linear time. Thus, it suffices to show that $\alpha$ is an optimal solution to the knapsack problem $(w, p, W)$, which is equivalent to the following two claims:

- Claim 1: We have $\sum_{i \in [n]} \alpha_i w_i \leq W$.
- Claim 2: For any $\alpha' \in \{0,1\}^n$ with $\sum_{i \in [n]} \alpha'_i w_i \leq W$, we have $\sum_{i \in [n]} \alpha'_i p_i \leq \sum_{i \in [n]} \alpha_i p_i$.

We now prove each claim as follows.

**Proof of Claim 1.** If $\alpha = (0, \ldots, 0)$, then the claim clearly holds. Suppose that $\alpha \neq (0, \ldots, 0)$. Then,

$$1 - \frac{1 + 2\sum_{i \in [n]} \alpha_i w_i}{\mathrm{size}(\mathsf{TS})} \geq \mathrm{ratio_{lo}}(\pi) \geq r \geq 1 - \frac{1 + 2W}{\mathrm{size}(\mathsf{TS})}.$$

Here the first inequality uses $\alpha \neq (0, \ldots, 0)$ and the definition of $\alpha$ and $\mathcal{M}$; the second inequality uses the fact that $\pi$ is a valid solution to the above tradeoff problem; and the third inequality uses the definition of $r$. Hence, the claim holds.

**Proof of Claim 2.** Suppose that the claim does not hold. Then, there exists $\alpha' \in \{0,1\}^n$ such that

$$\sum_{i \in [n]} \alpha'_i w_i \leq W, \qquad \sum_{i \in [n]} \alpha'_i p_i > \sum_{i \in [n]} \alpha_i p_i.$$

Define a precision assignment $\pi' \in \Pi(\mathcal{C})$ as:

$$\pi'(dv_{2n+1}) \triangleq \mathsf{fp_{hi}},$$
$$\pi'(d\theta_i) \triangleq \pi'(dv_{n+i}) \triangleq \mathsf{fp_{hi}} \quad \text{for all } i \in [n] \text{ with } \alpha'_i = 1,$$
$$\pi'(t) \triangleq \mathsf{fp_{lo}} \quad \text{for all other } t \in \mathsf{TS}.$$

Then, we have $\mathrm{ratio_{lo}}(\pi') \geq r$ by $\sum_{i \in [n]} \alpha'_i w_i \leq W$ and the definition of $\pi'$, $\mathcal{M}$, and $r$. Hence, it suffices to show $\mathrm{acc}(\pi) < \mathrm{acc}(\pi')$, because this would contradict to the fact that $\pi$ is an optimal solution.

To show $\mathrm{acc}(\pi) < \mathrm{acc}(\pi')$, we prove the following two lemmas: the first lemma gives a closed form of $\mathrm{acc}(\pi)$ and $\mathrm{acc}(\pi')$, and the second lemma shows that $\sum_{i \in [n]} \beta_i w_i x_i^2$ is close to $\sum_{i \in [n]} \beta_i p_i$ (where the former summation appears in $\mathrm{acc}(\pi)$ and $\mathrm{acc}(\pi')$).

*Lemma A.1. The following hold:*

$$\mathrm{acc}(\pi) = 2^{-k}y + 2^{-(2k+l)}\sum_{i \in [n]} \alpha_i w_i x_i^2, \qquad \mathrm{acc}(\pi') = 2^{-k}y + 2^{-(2k+l)}\sum_{i \in [n]} \alpha'_i w_i x_i^2.$$

*Proof.* We prove the equation for $\mathrm{acc}(\pi)$ only, since the equation for $\mathrm{acc}(\pi')$ can be proved similarly.

First, we show that for all $i \in [n]$ and $j \in [w_i]$,

$$\hat{d}\theta_{i,j} = \alpha_i \cdot 2^{-k} x_i. \tag{7}$$

Pick any $i \in [n]$ and $j \in [w_i]$. Note that by the definition of $\mathcal{M}$, we have

$$\hat{d}\theta_{i,j} = \mathrm{rnd}_{\pi(d\theta_i)}\Big(\mathrm{rnd}_{\pi(dv_{n+i})}(\mathrm{rnd}_{\pi(dv_{2n+1})}(2^{-k})) \cdot \mathrm{rnd}_{v_i}(\mathrm{rnd}_{v_0}(x_i))\Big)$$
$$= \mathrm{rnd}_{\pi(d\theta_i)}\Big(\mathrm{rnd}_{\pi(dv_{n+i})}(\mathrm{rnd}_{\pi(dv_{2n+1})}(2^{-k})) \cdot x_i\Big),$$

where the second equality uses Eq. (3) and that $x_i$ is representable in $\mathsf{fp_{lo}}$. We prove Eq. (7) by case analysis on $\alpha_i$. Suppose $\alpha_i = 1$. Then, by the definition of $\alpha_i$, $\pi(d\theta_i) = \pi(dv_{n+i}) = \pi(dv_{2n+1}) = \mathsf{fp_{hi}}$. From this, we get the desired equation:

$$\hat{d}\theta_{i,j} = \mathrm{rnd_{hi}}\Big(\mathrm{rnd_{hi}}(\mathrm{rnd_{hi}}(2^{-k})) \cdot x_i\Big) = \mathrm{rnd_{hi}}(2^{-k} \cdot x_i) = 2^{-k}x_i,$$

where the last two equalities use Eq. (6). Suppose now $\alpha_i = 0$. Then, by the definition of $\alpha_i$, at least one of $\pi(d\theta_i)$, $\pi(dv_{n+i})$, and $\pi(dv_{2n+1})$ is $\mathsf{fp_{lo}}$. If $\pi(dv_{n+i}) = \mathsf{fp_{lo}}$ or $\pi(dv_{2n+1}) = \mathsf{fp_{lo}}$, we get the desired equation:

$$\hat{d}\theta_{i,j} = \mathrm{rnd}_{\pi(d\theta_i)}\Big(\mathrm{rnd_{lo}}(2^{-k}) \cdot x_i\Big) = \mathrm{rnd}_{\pi(d\theta_i)}(0 \cdot x_i) = 0,$$

where the first equality uses Eq. (3) and Eq. (6), and the second equality uses Eq. (5). The remaining case is when $\pi(dv_{n+i}) = \pi(dv_{2n+1}) = \mathsf{fp_{hi}}$ and $\pi(d\theta_i) = \mathsf{fp_{lo}}$. We get the desired equation in this case as well:

$$\hat{d}\theta_{i,j} = \mathrm{rnd_{lo}}\Big(\mathrm{rnd_{hi}}(\mathrm{rnd_{hi}}(2^{-k})) \cdot x_i\Big) = \mathrm{rnd_{lo}}(2^{-k} \cdot x_i) = 0,$$

where the second equality uses Eq. (6), and the last equality uses Eq. (5). Hence, we have proved Eq. (7).

Next, let $\theta_i$ be the $i$-th parameter tensor before training starts, and $\theta'_i$ be the corresponding tensor after training ends ($i \in [n]$). Then, by the definition of the tradeoff problem constructed above, we have $\theta_{i,j} = 0$ and

$$\theta'_{i,j} = \theta_{i,j} - \mathrm{rnd_{hi}}(2^{-l} \cdot \hat{d}\theta_{i,j}) = 0 - \mathrm{rnd_{hi}}(2^{-l} \cdot (\alpha_i \cdot 2^{-k}x_i)) = \alpha_i \cdot (-2^{-(k+l)}x_i),$$

where the second equality uses Eq. (7) and the third equality uses Eq. (6). Using this equation, we finally obtain the conclusion of this lemma:

$$
\begin{aligned}
\mathrm{acc}(\pi) &= -L(f_{\theta'}(x), y) \\
&= -2^{-k}\Big| y - \sum_{i \in [n]} \sum_{j \in [w_i]} \theta'_{i,j} x_i \Big| \\
&= -2^{-k}\Big| y - \sum_{i \in [n]} \sum_{j \in [w_i]} \alpha_i \cdot (-2^{-(k+l)} x_i) \cdot x_i \Big| \\
&= -2^{-k}\Big| y + \sum_{i \in [n]} \alpha_i \cdot 2^{-(k+l)} w_i x_i^2 \Big| \\
&= 2^{-k}\Big( y + \sum_{i \in [n]} \alpha_i \cdot 2^{-(k+l)} w_i x_i^2 \Big) \\
&= 2^{-k} y + 2^{-(2k+l)} \sum_{i \in [n]} \alpha_i w_i x_i^2,
\end{aligned}
$$

where the first two equalities use the definition of accuracy, and the second last equality uses the definition of $y$. This concludes the proof of the lemma. ∎

*Lemma* A.2. *For any $\beta \in \{0,1\}^n$,*

$$
\Big| \sum_{i \in [n]} \beta_i w_i x_i^2 - \sum_{i \in [n]} \beta_i p_i \Big| < \frac{1}{2}.
$$

*Proof.* We first show that for any $i \in [n]$,

$$
|w_i x_i^2 - p_i| < \frac{1}{2n}.
$$

Pick any $i \in [n]$. By Eq. (4) and the definition of $x_i$, we have

$$
\Big| x_i - \sqrt{\frac{p_i}{w_i}} \Big| < \sqrt{\frac{p_i}{w_i}} \cdot \frac{1}{6n \cdot \max_{j \in [n]} p_j} \le \sqrt{\frac{p_i}{w_i}} \cdot \frac{1}{6np_i}.
$$

From this, we have

$$
\sqrt{\frac{p_i}{w_i}}\Big( 1 - \frac{1}{6np_i} \Big) < x_i < \sqrt{\frac{p_i}{w_i}}\Big( 1 + \frac{1}{6np_i} \Big), \qquad \frac{p_i}{w_i}\Big( 1 - \frac{1}{6np_i} \Big)^2 < x_i^2 < \frac{p_i}{w_i}\Big( 1 + \frac{1}{6np_i} \Big)^2.
$$

From this, we obtain the desired result:

$$
|w_i x_i^2 - p_i| < p_i\Big( \Big( 1 + \frac{1}{6np_i} \Big)^2 - 1 \Big) = p_i\Big( \frac{1}{3np_i} + \frac{1}{(6np_i)^2} \Big) < p_i\Big( \frac{1}{3np_i} + \frac{1}{6np_i} \Big) = p_i \cdot \frac{1}{2np_i} = \frac{1}{2n},
$$

where the second inequality uses $6np_i > 1$ (as $n, p_i \in \mathbb{N}$).

Using this result, we can show the conclusion as follows:

$$
\Big| \sum_{i \in [n]} \beta_i w_i x_i^2 - \sum_{i \in [n]} \beta_i p_i \Big| = \Big| \sum_{i \in [n]} \beta_i (w_i x_i^2 - p_i) \Big| \le \sum_{i \in [n]} |\beta_i| \cdot |w_i x_i^2 - p_i| < \sum_{i \in [n]} \frac{1}{2n} = \frac{1}{2},
$$

where the last inequality uses $|\beta_i| \le 1$. This completes the proof of the lemma. ∎

Using the two lemmas, we now prove $\mathrm{acc}(\pi) < \mathrm{acc}(\pi')$. By Lemma A.2 and $\sum_{i \in [n]} \alpha_i p_i < \sum_{i \in [n]} \alpha'_i p_i$, we have

$$
\sum_{i \in [n]} \alpha_i w_i x_i^2 < \sum_{i \in [n]} \alpha_i p_i + \frac{1}{2} \le \sum_{i \in [n]} \alpha'_i p_i - \frac{1}{2} < \sum_{i \in [n]} \alpha'_i w_i x_i^2,
$$

where the second inequality comes from $\alpha_i, \alpha'_i \in \{0,1\}$ and $p_i \in \mathbb{N}$. From this, and by Lemma A.1, we obtain $\mathrm{acc}(\pi) < \mathrm{acc}(\pi')$ as desired. This concludes the proof of Claim 2, thereby finishing the proof of the theorem. □

**Remark A.3.** In the proof of Theorem 3.2, we proved the NP-hardness of Problem 3.1 by making use of only a few limited aspects of the problem. For instance, we used the fact that some values representable in a high-precision format round to *zero* in a low-precision format; on the other hand, many other values representable in a high-precision format round to *non-zero* values in a low-precision format, and this indeed occurs in practical training (even more frequently than underflows). Also, we used a simple setting for training in which a gradient descent algorithm is applied for *one epoch*, training data consist of *one input-output pair*, and test data is *the same as* training data; on the other hand, in practical training, a gradient descent algorithm is applied for *many epochs*, training data consists of *many input-output pairs*, and test data is *different from* training data.

Problem 3.1 is general enough so that it embraces all the aforementioned aspects of floating-points and training, including those that are not considered in the proof of Theorem 3.2. Since those aspects are likely to make the problem even more difficult, we conjecture that the problem would be more intractable than being NP-hard. □

## B  Experiments: Deferred Details

The datasets we use have the following licenses:

- CIFAR-10 and CIFAR-100: These datasets are under the MIT license.
- ImageNet: This dataset can be used "only for non-commercial research and educational purposes." For more details, see its webpage (Stanford Vision Lab, 2020).

The implementations of models we use have the following licenses:

- SqueezeNet for CIFAR-10 and CIFAR-100: We adapt an implementation of the model in a public GitHub repository (Pathak, 2020), whose license information is not available.
- ShuffleNet-v2, MobileNet-v2, and ResNet-18 for CIFAR-10 and CIFAR-100: We adapt an implementation of these models in a public GitHub repository (kuangliu, 2021), which is under the MIT license.
- ShuffleNet-v2 for ImageNet and ImageNet-200-$i$: We adapt an implementation of the model in the torchvision library (PyTorch, 2022b), which is under the BSD 3-Clause license.

The details of how we train models are as follows:

- Four models on CIFAR-10 and CIFAR-100: We train the four models with a standard setup (kuangliu, 2021). In particular, we run the (non-Nesterov) SGD optimizer for 200 epochs with minibatch size of 128 (over 1 GPU), learning rate of 0.1, momentum of 0.9, weight decay of $5 \times 10^{-4}$, and the cosine annealing scheduler for learning rate. For dynamic loss scaling, we use initial scale of $2^{16}$, growth factor of 2, back-off factor of 0.5, and growth interval of 1 epoch, as suggested in PyTorch (PyTorch, 2022a).
- ShuffleNet-v2 on ImageNet: We train the model with the default setup given in PyTorch's GitHub repository (PyTorch, 2022c), except that we use larger minibatch size and learning rate as in (Goyal et al., 2017; Kalamkar et al., 2019; Krizhevsky, 2014; PyTorch, 2022d) to reduce the wall-clock time of training. In particular, we run the (non-Nesterov) SGD optimizer for 90 epochs with minibatch size of 1024 (over 16 GPUs), learning rate of 0.4, momentum of 0.9, weight decay of $10^{-4}$, and the cosine annealing scheduler for learning rate. For dynamic loss scale, we use initial scale of $2^{16}$, growth factor of 2, back-off factor of 0.5, and growth interval of 0.5 epoch, as suggested in PyTorch (PyTorch, 2022a).
- ShuffleNet-v2 on ImageNet-200-$i$: We train the model with the same settings for ImageNet except that we use the default values for minibatch size and learning rate given in (PyTorch, 2022c), i.e., minibatch size of 256 (over 4 GPUs) and learning rate of 0.1.

## C  Experiments: Deferred Results

### C.1  Comparison with Existing Precision Assignments

Figure 9 presents a zoomed-in version of Figure 3 (left).

Figure 10 presents results omitted in Figure 4: training results of smaller variant models (which have width multiplier 0.5 or 0.1) on CIFAR-100 with $\pi_{\mathsf{fp32}}$, $\pi_{\mathrm{unif}}$, $\pi_{\mathrm{op}}$, $\pi_{\mathrm{op}'}$, and $\pi_{\mathrm{ours},r}$. The figure shows similar results

to Figure 4: the results for the variant models with width multiplier 0.5 (and 0.1) are similar to those for the original models (and the variant models with width multiplier 0.25).

Figures 11 and 12 show the average training trajectories for the configurations presented in Figures 4 and 10.

Figures 13 and 14 present the same results as Figures 3 and 4 except the following: in the former, $\pi_{\text{op}}$ and $\pi_{\text{op}'}$ are equipped with our precision promotion technique, whereas in the latter they do not so. Figures 13 and 14 do not include $\pi_{\text{unif}}$ because this assignment with the precision promotion is identical to $\pi_{\text{ours},1}$.

### C.2 Ablation Study: Precision Demotion and Promotion

Figure 16 presents results omitted in Figure 5: training results of ResNet-18 on CIFAR-100 with $\pi_{\text{ours},r}$, $\pi_{\text{ours[inc]},r}$, and $\pi_{\text{ours[rand]},r}$. The figure shows similar results to Figure 5 except that it shows smaller differences in memory-accuracy tradeoff between the three precision assignments.

Figure 17 presents results omitted in Figure 6: training results of four models on CIFAR-10 with $\pi_{\text{ours},r}$ and $\pi_{\text{ours[no-promo]},r}$. The figure shows similar results to Figure 6 except that the training of ResNet-18 on CIFAR-10 does not diverge even with $\pi_{\text{ours[no-promo]},r}$ for all $r$ values.

## References for Appendix

Priya Goyal, Piotr Dollár, Ross B. Girshick, Pieter Noordhuis, Lukasz Wesolowski, Aapo Kyrola, Andrew Tulloch, Yangqing Jia, and Kaiming He. Accurate, Large Minibatch SGD: Training ImageNet in 1 Hour. *arXiv:1706.02677*, 2017.

Dhiraj D. Kalamkar, Dheevatsa Mudigere, Naveen Mellempudi, Dipankar Das, Kunal Banerjee, Sasikanth Avancha, Dharma Teja Vooturi, Nataraj Jammalamadaka, Jianyu Huang, Hector Yuen, Jiyan Yang, Jongsoo Park, Alexander Heinecke, Evangelos Georganas, Sudarshan Srinivasan, Abhisek Kundu, Misha Smelyanskiy, Bharat Kaul, and Pradeep Dubey. A Study of BFLOAT16 for Deep Learning Training. *arXiv:1905.12322*, 2019.

Richard M. Karp. Reducibility Among Combinatorial Problems. In *Complexity of Computer Computations*, pp. 85–103, 1972.

Alex Krizhevsky. One weird trick for parallelizing convolutional neural networks. *arXiv:1404.5997*, 2014.

kuangliu. https://github.com/kuangliu/pytorch-cifar, 2021.

Gaurav Pathak. https://github.com/gsp-27/pytorch_Squeezenet, 2020.

PyTorch. Documentation of `torch.amp`. https://pytorch.org/docs/stable/amp.html, 2022a.

PyTorch. https://github.com/pytorch/vision/tree/main/torchvision/models, 2022b.

PyTorch. https://github.com/pytorch/vision/tree/main/references/classification, 2022c.

PyTorch. https://github.com/pytorch/vision/tree/main/references/classification#resnext, 2022d.

Stanford Vision Lab. https://image-net.org/download.php, 2020.

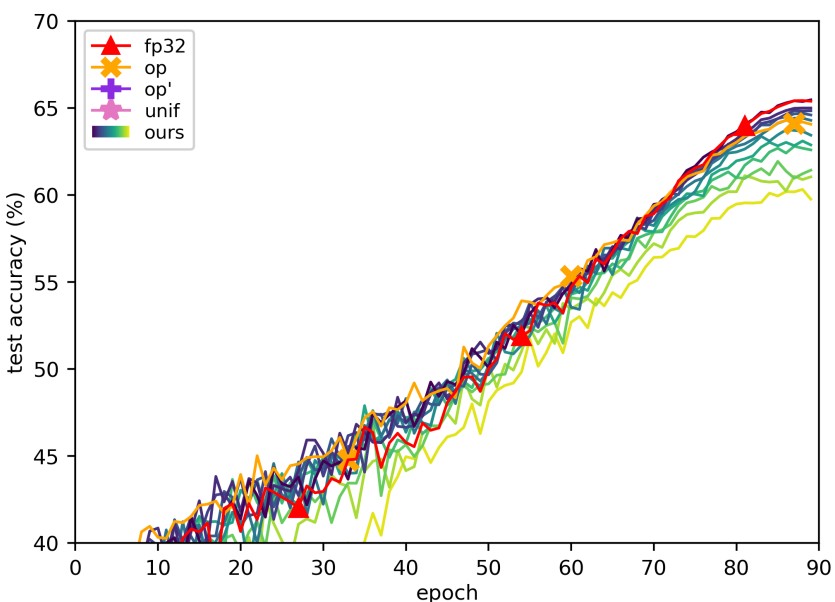

Figure 9: A zoomed-in version of Figure 3 (left). Results of training ShuffleNet-v2 on ImageNet with $\pi_{\text{fp32}}$, $\pi_{\text{unif}}$ (Micikevicius et al., 2018), $\pi_{\text{op}}$ (Sun et al., 2019), $\pi_{\text{op}'}$ (Kalamkar et al., 2019), and $\pi_{\text{ours},r}$. Each line shows the average training trajectory for each precision assignment; $\pi_{\text{ours},r}$ is colored from navy to yellow (darker for smaller $r$).

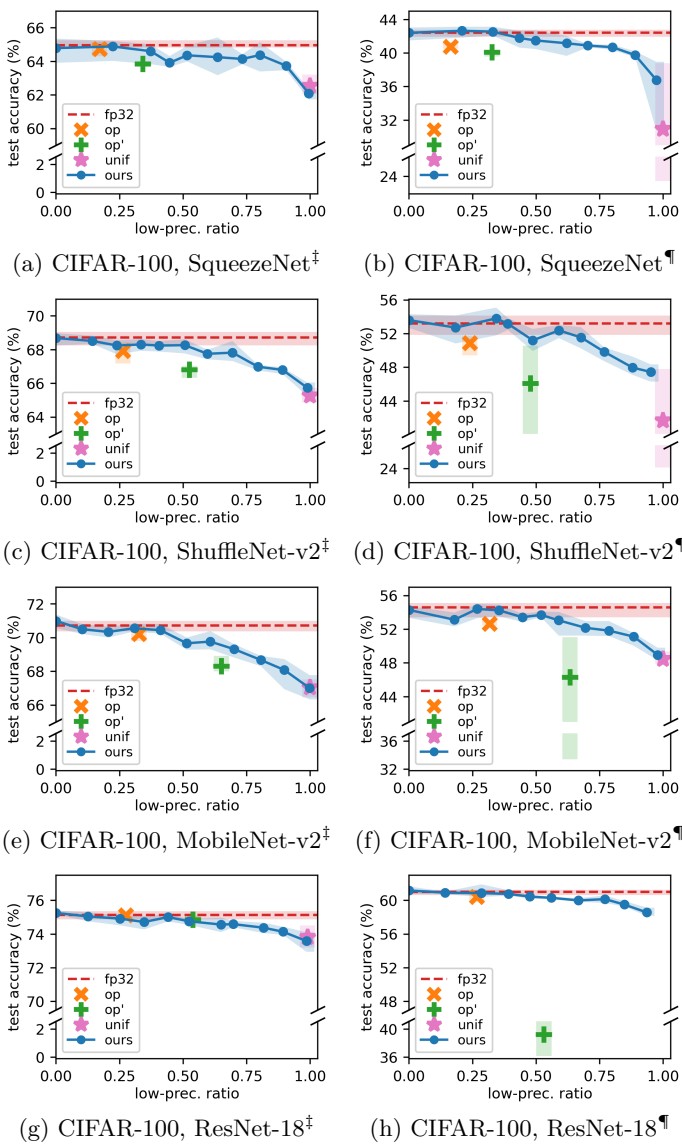

Figure 10: Continued from Figure 4. Memory-accuracy tradeoffs of $\pi_{\text{unif}}$ (Micikevicius et al., 2018), $\pi_{\text{op}}$ (Sun et al., 2019), $\pi_{\text{op}'}$ (Kalamkar et al., 2019), and $\pi_{\text{ours},r}$ for smaller variants of four models on CIFAR-100. The variant models have width multiplier 0.5 (marked by [‡]) or 0.1 (marked by [¶]). Top-right points are better than bottom-left ones. In all but one plots, there are ●s above and to the right of ✖ and ✚, respectively; even in the one plot (g), ●s have almost the same tradeoffs to ✖ and ✚. In three of all plots, ★ has much smaller y-values than other points; ★ is missing in (h) as its y-value is too small.

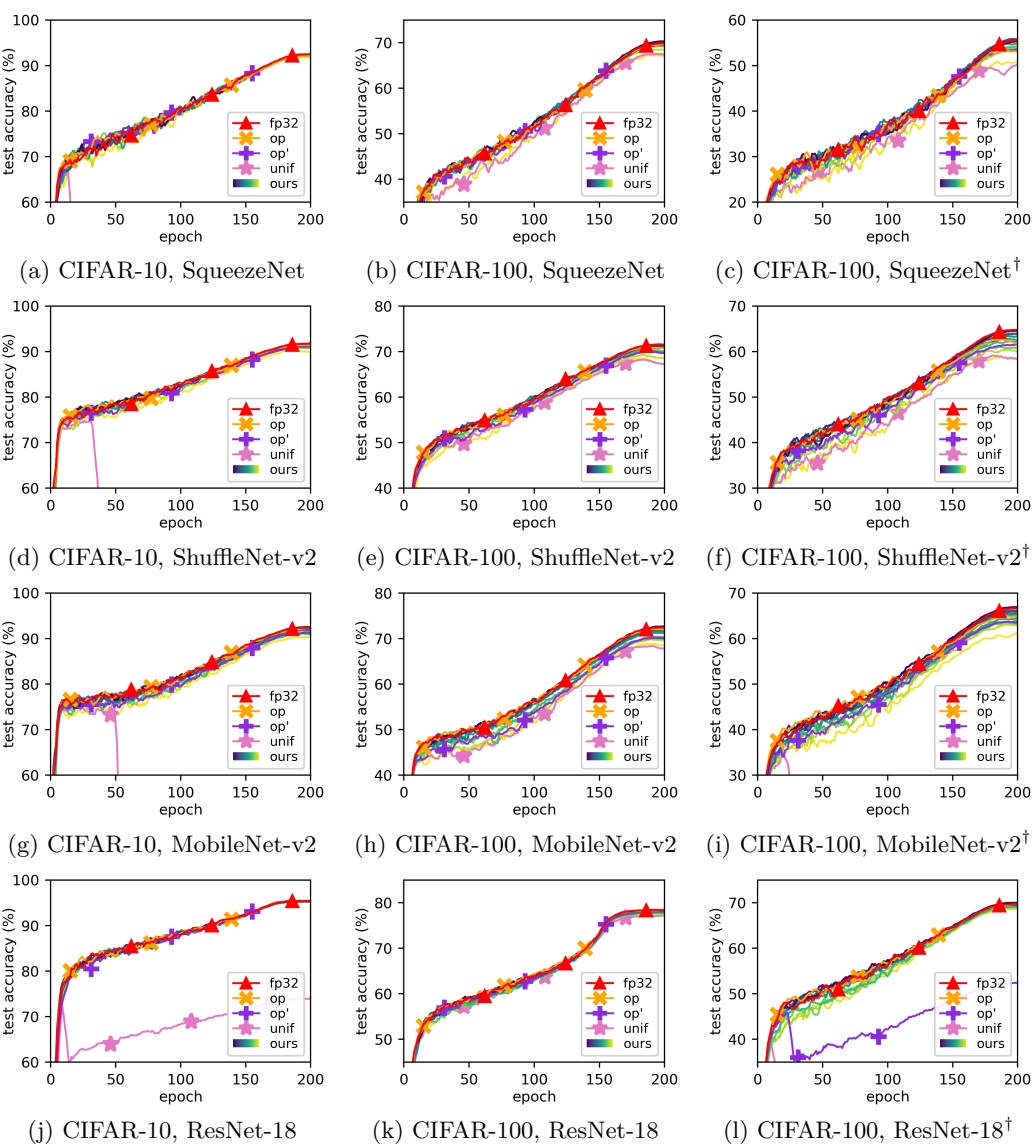

Figure 11: Training trajectories for the configurations shown in Figure 4. Each line shows the average training trajectory for each precision assignment. $\pi_{\mathrm{ours},r}$ is colored from navy to yellow (darker for smaller $r$).

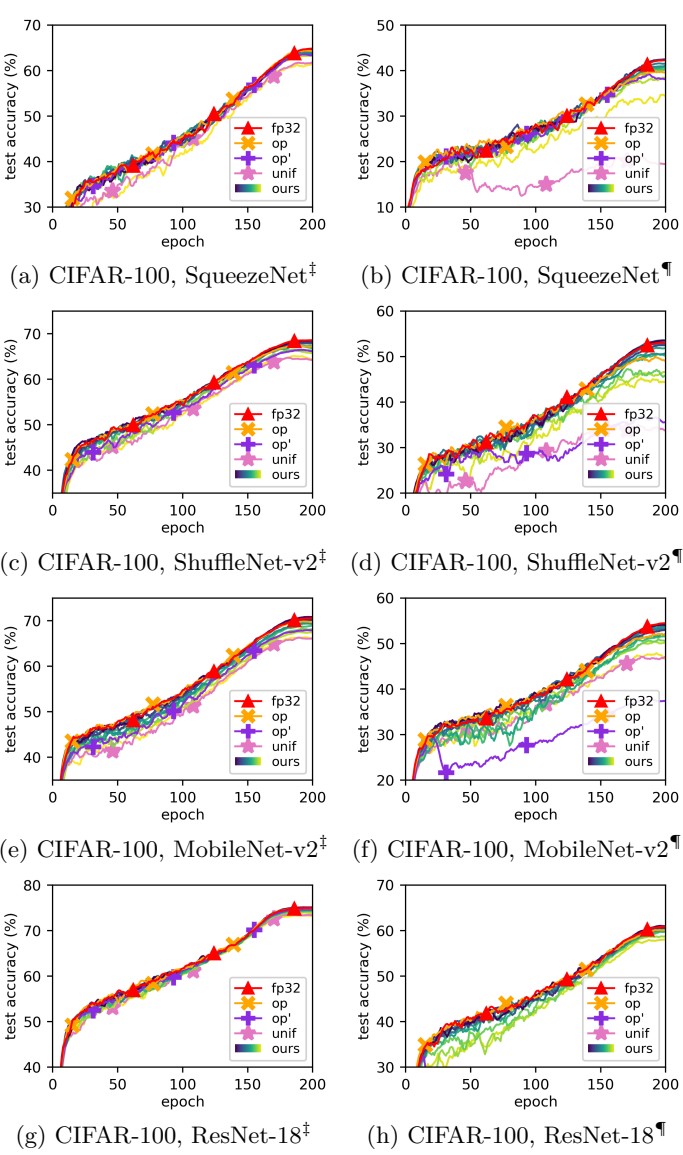

(a) CIFAR-100, SqueezeNet$^{\ddagger}$    (b) CIFAR-100, SqueezeNet$^{\P}$

(c) CIFAR-100, ShuffleNet-v2$^{\ddagger}$    (d) CIFAR-100, ShuffleNet-v2$^{\P}$

(e) CIFAR-100, MobileNet-v2$^{\ddagger}$    (f) CIFAR-100, MobileNet-v2$^{\P}$

(g) CIFAR-100, ResNet-18$^{\ddagger}$    (h) CIFAR-100, ResNet-18$^{\P}$

Figure 12: Training trajectories for the configurations shown in Figure 10. Each line shows the average training trajectory for each precision assignment. $\pi_{\mathrm{ours},r}$ is colored from navy to yellow (darker for smaller $r$).

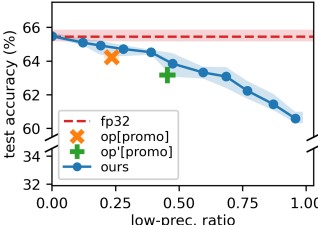

Figure 13: Results corresponding to Figure 3. The only difference from Figure 3 is that $\pi_{\text{op}}$ and $\pi_{\text{op}'}$ here are equipped with our precision promotion technique, whereas $\pi_{\text{op}}$ and $\pi_{\text{op}'}$ in the previous figure do not so.

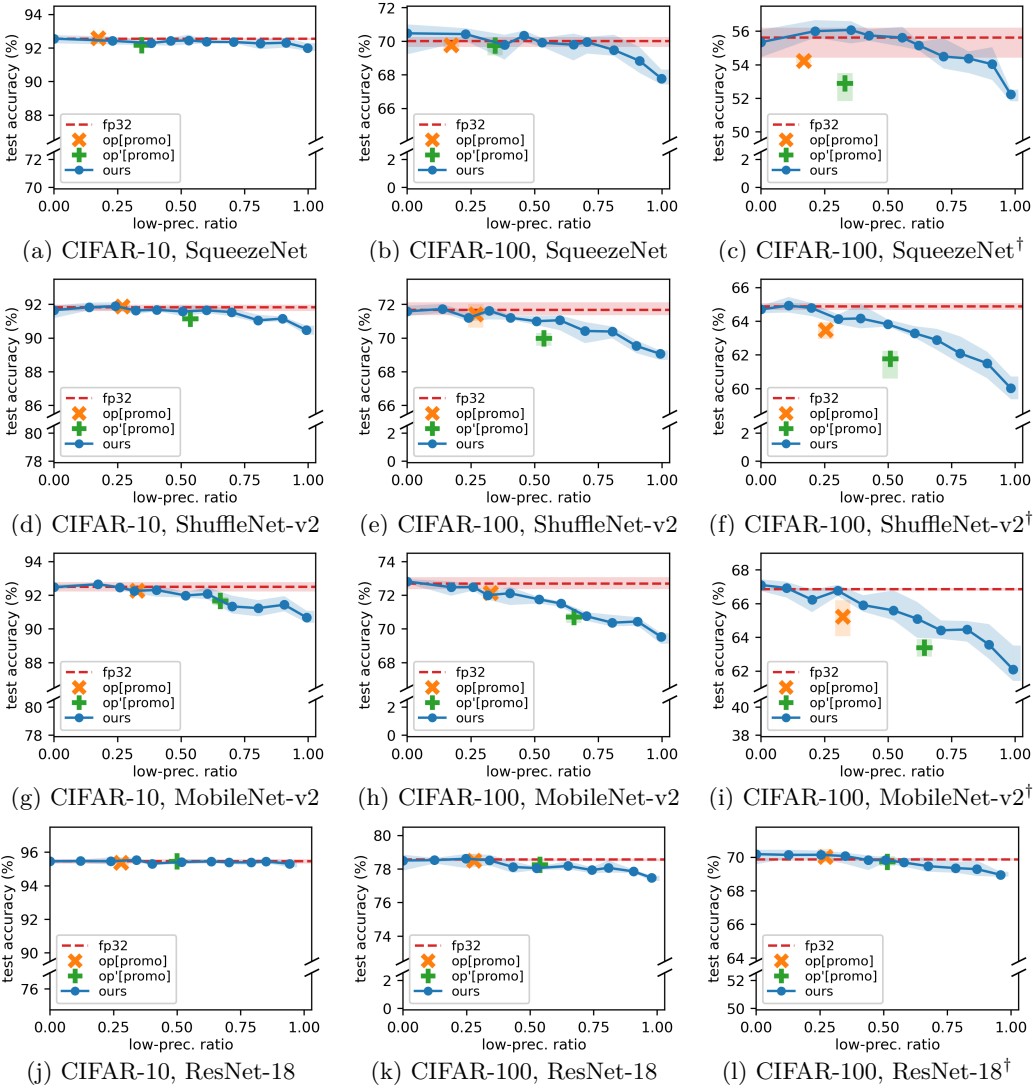

Figure 14: Results corresponding to Figure 4. The only difference from Figure 4 is that $\pi_{\text{op}}$ and $\pi_{\text{op}'}$ here are equipped with our precision promotion technique, whereas $\pi_{\text{op}}$ and $\pi_{\text{op}'}$ in the previous figure do not so.

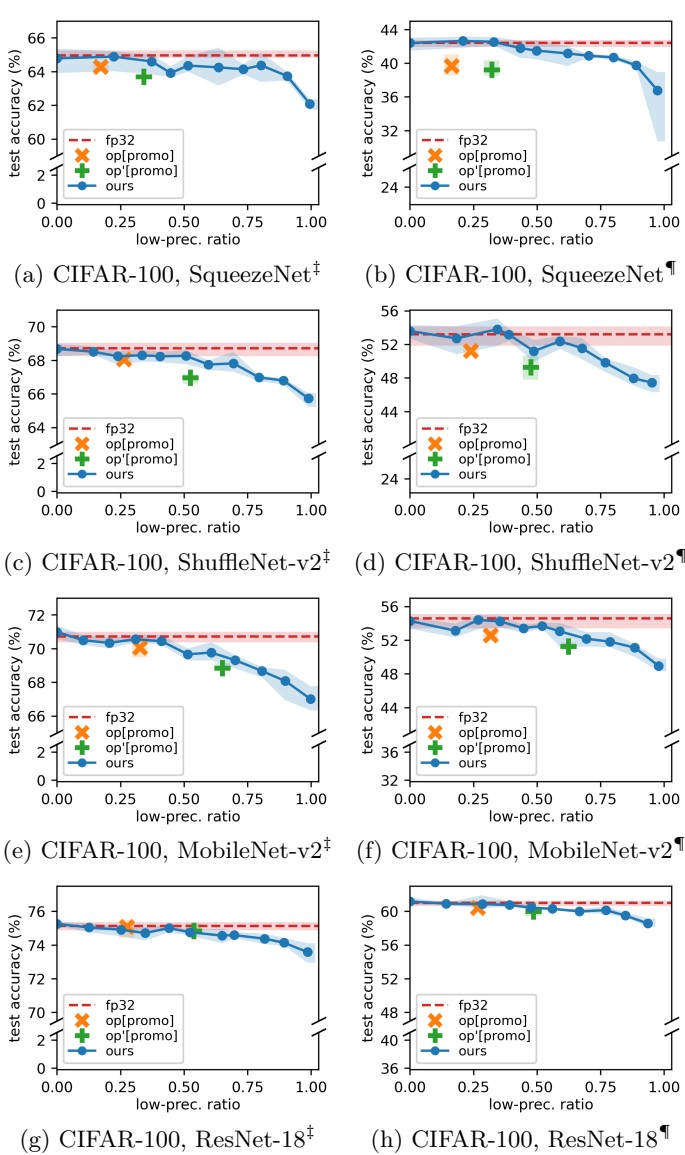

(a) CIFAR-100, SqueezeNet[‡]    (b) CIFAR-100, SqueezeNet[¶]

(c) CIFAR-100, ShuffleNet-v2[‡]    (d) CIFAR-100, ShuffleNet-v2[¶]

(e) CIFAR-100, MobileNet-v2[‡]    (f) CIFAR-100, MobileNet-v2[¶]

(g) CIFAR-100, ResNet-18[‡]    (h) CIFAR-100, ResNet-18[¶]

Figure 15: Results corresponding to Figure 10. The only difference from Figure 10 is that $\pi_{\mathrm{op}}$ and $\pi_{\mathrm{op'}}$ here are equipped with our precision promotion technique, whereas $\pi_{\mathrm{op}}$ and $\pi_{\mathrm{op'}}$ in the previous figure do not so.

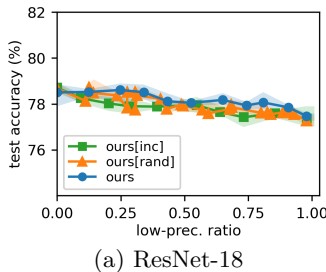

(a) ResNet-18

Figure 16: Continued from Figure 5. Memory-accuracy tradeoffs of $\pi_{\text{ours},r}$, $\pi_{\text{ours[inc]},r}$, and $\pi_{\text{ours[rand]},r}$ for ResNet-18 on CIFAR-100. Observe that ●s are above and to the right of other points in nearly all cases.

(a) CIFAR-10, SqueezeNet

(b) CIFAR-10, ShuffleNet-v2

(c) CIFAR-10, MobileNet-v2

(d) CIFAR-10, ResNet-18

Figure 17: Continued from Figure 6. Training four models on CIFAR-10 with $\pi_{\text{ours},r}$ and $\pi_{\text{ours[no-promo]},r}$. Column 1: Training trajectories of $\pi_{\text{ours[no-promo]},r}$ for different $r$; colors denote $r$ values (darker for smaller $r$). Column 2: Top-5 overflow ratios of tensors at each epoch, for the highlighted trajectory in (a); the largest ratio is blue and the fifth largest red. Column 3: Memory-accuracy tradeoffs of $\pi_{\text{ours},r}$ and $\pi_{\text{ours[no-promo]},r}$. Column 4: Low-precision ratio when training ends vs. when training starts, for $\pi_{\text{ours},r}$ and $\pi_{\text{ours[no-promo]},r}$.

