# OpenReview forum: "Training with Mixed-Precision Floating-Point Assignments"
_TMLR — Accepted by TMLR_

### Review · Reviewer_eNE7 · 2023-04-10

**Summary Of Contributions:**

**Summary**

This work studies mixed-precision training in 8/16-bit with the goal of performing as many operations as possible in 8-bit Floats without losing accuracy. This is achieved by using a system of promotion and demotion to and from high-precision as well as a heuristic to initialize the tensors so that a given ratio uses 8-bit tensors.

**Contributions**:
- Develop an algorithm for promotion/demotion and compare it to uniform precision assignments
- Developing an algorithm how for assigning the precision to initial tensors
- Studying and analyzing the trade-offs between

**Audience:**

Yes

**Broader Impact Concerns:**

Currently, the work has no Broader Impact Statement section. I think that is okay. One could make an argument that better FP8 training makes it easier to train more powerful networks with fewer resources. This can have an impact for people with the least resources to make more powerful models available, but it can also help large corporations to train even more powerful networks for the same costs. Both of these downstream effects would affect society. As such, a Broader Impact Statement could be warranted.

**Claims And Evidence:**

No

**Requested Changes:**

I think in the current form, your claims are too general.

I think the following adjustments to your claims or additional experiments to support your claims would be suitable:
- Your method is currently only shown for convolutional networks. Either highlight that your method works just for convolutional networks, or add transformer experiments. Additional experiments that would convince me that your method is more general would be to repeat your experimental method but to use functions that your software currently does not support in 16/32-bit (such as softmax). Then use this setup and train either a CLIP model (OpenCLIP) or text-only transformer on WikiText-103.
- I think your claim that you outperform existing precision assignments is not warranted. I would see "all GEMM in low-precision, and everything else in high-precision" also as a heuristic not unlike your, but currently, you do not compare against it. This heuristic has been very successful (Micikevicius et al., 2022, FP8 Formats for Deep Learning), but it is known to fail in some large-scale scenarios. Otherwise, you could reduce your claims and say you outperform methods where at least some non-GEMM operations are done in FP8.

I deems these claim-reductions or additional experiments critical for acceptance since the more general claims are not supported by the evidence in the paper.

**Strengths And Weaknesses:**

### Strengths:
- Careful study of how the variables related to the mixed-precision assignment algorithm affect performance
- Careful comparison with baselines

### Weaknesses
- The predictive performance of the method is relatively poor. This is because the authors perform intermediate operations between GEMM operations in FP8 if his group has been demoted. A better practice would be to perform just GEMM operations in low precision because 95% of parameters and computation are done in these layers; other layers have little benefit from FP8 computation.
- No experimental results on transformers. The authors claim that it is difficult to implement their method for transformers and thus cannot be studied easily. Asides from the softmax operation (which is usually done in FP32 anyways), there is no operation in transformers that is inherently different from convolutional networks (except convolution, but this can be seen as a matrix multiplication too).
- No experimental results at scale. Recent literature highlights that many successful low-precision methods fail at scale. Larger models introduce new problems and it is unclear if the authors algorithm can work at scale.

---

> ### Author Response · Authors · 2023-05-02
> **Thanks for the review**
>
> We greatly appreciate your thoughtful comments. We address your concerns as follows.
>
> ### **Highlight that our method works just for convolutional networks.**
>
> To highlight this point, we have added text to the abstraction and introduction (pages 1-3).
>
> ### **There is no comparison against “all GEMM in low-precision and everything else in high-precision”.**
>
> We clarify that the precision assignment used in [Micikevicius+2022] (i.e., “all GEMM in low-precision and everything else in high-precision”) is the *operator-based assignment* $\pi_{op}$ considered in our paper: it is formally defined and informally explained in our paper (see, e.g., Eq. (2) and surrounding text in Section 3.1). In our experiments (Section 5), we did compare our precision assignments against this assignment $\pi_{op}$ (and others as well).

---

> > ### Comment · Reviewer_eNE7 · 2023-05-21
> > **Need clarification before I write the review.**
> >
> > Sorry for being late with my response, but before I can evaluate your updated submission, I need to understand your results with the operator-based assignment. I would expect operator-based assignment to be close to 99% of low-precision ratio, since biases and batch norms usually make less than 1% of parameters, and these are the only operations that would be done in high-precision. But your plots show the ratio to be between 0.25 and 0.33. Could you explain this?
> >
> > I think otherwise, you still need to reduce your claims. You still claim:
> > >We provide a technique that explores this memory-accuracy tradeoff by generating precision assignments that (i)
> > use less memory and (ii) lead to more accurate models at the same time, compared to the preci-
> > sion assignments considered by prior work in low-precision floating-point training.
> >
> > The added passage after this claim, does not reduce it:
> > >We evaluate our technique on image classification tasks by training convolutional networks on CIFAR-10,
> > CIFAR-100, and ImageNet.
> >
> > The correct claim would be:
> > "We provide a technique that explores this memory-accuracy tradeoff for **convolutional neural networks** by generating precision assignments that (i)
> > use less memory and (ii) lead to more accurate **convolutional networks** at the same time, compared to the preci-
> > sion assignments considered by prior work in low-precision floating-point training.
> >
> > Reducing the claims for me means not having a general claim, but the specific claim that is supported by your data.

---

> > > ### Author Response · Authors · 2023-05-25
> > > **Thanks for the additional comments.**
> > >
> > > We appreciate your additional comments.
> > >
> > > ### **Clarification on our results**
> > >
> > > We clarify that the low-precision ratio is about **all tensors** appearing in a gradient computation, not about only **parameter tensors** used in the computation. More precisely, the low-precision ratio of $\pi$ is defined as “the portion of the tensors represented in low-precision under $\pi$, among all tensors appearing in a gradient computation” (page 6).
> > >
> > > As you mentioned, the low-precision ratio of the operator-based assignment $\pi_{op}$ is indeed between 0.25 and 0.35 in our experiments. This follows from the fact that the assignment uses low precision only in the input tensors of GEMM operators (in both forward and backward pass). You can see that the low-precision ratio of $\pi_{op’}$ (i.e., a variant of $\pi_{op}$) is roughly two times that of $\pi_{op}$. This follows from the fact that $\pi_{op’}$ uses low precision in both the input and output tensors of GEMM operators.
> > >
> > > ### **Reducing our claims**
> > >
> > > Thank you for the additional suggestions. We agree that these suggestions would further clarify our contributions. We have incorporated them into our draft and uploaded it in the above.

---

### Review · Reviewer_LPEm · 2023-04-16

**Summary Of Contributions:**

This work targets the problem of mapping from all tensors (arising in training) to precision levels (high or low) in DNN training. Specifically, the authors start with the largest tensor in the model, assign tensors to low precision in size order (from largest to smallest) until the model aggregate falls below the given upper bound, and keep all remaining tensors high precision. The experiments are conducted on CIFAR-10/100 and ImageNet with models as SqueezeNet, ShuffleNet-v2, MobileNet-v2, and ResNet-18.

**Audience:**

Yes

**Broader Impact Concerns:**

No concern about the ethical implications.

**Claims And Evidence:**

Yes

**Requested Changes:**

Please see #Weakness above.

**Strengths And Weaknesses:**

# Strengths
> + Well-motivated: using low-precision tensor in DNN training is a common technique to reduce the memory requirement and training time. However, using one precision setting for all the tensors may not be the optimal solution. It is important to design a mapping function to map different tensors to different precision.
> Idea simple and effective: assigning tensors to low precision in size order is simple and can be supported by most devices and even pre-computed offline and the experiments validate that such a simple scheme can beat existing precision assignment methods in both commonly used models and compact models.
> + Comprehensive ablation studies: the ablation studies cover both the studies on each single technique and hyperparameters.

# Weakness
> + Limited tasks: all the tasks are done in image classification tasks. Since the proposed method is a heuristic and greedy approach and the improvement over the previous precision assignment is not so large, it would be better to show this method is generally applicable to different tasks.
> + Unclear hardware benefit: the authors claim that the selection of precision options (e.g., FP8 or FP32) is to support the incoming hardware. However, memory saving is measured by a self-defined low-precision ratio. Although I fully understand the hardware measurement may be difficult now because of the limited access to the new hardware, but a low-precision ratio may be too simple to evaluate the memory cost. For example, because of the proposed Precision Promotion, every tensor has the opportunity to be high-precision. Then the hardware has to design sufficient memory for such cases. Thus, memory utilization can be low and the real memory saving may be smaller than the low-precision ratio because of the cases that low-precision tensor can be high-precision and extra cost to the data type casting.
> +  More straightforward illustration of the proposed algorithm: it would be better to add a diagram or an algorithm table to show the proposed algorithm to avoid the cases that readers lose in the formulas in Sec. 4.

---

> ### Author Response · Authors · 2023-05-02
> **Thanks for the review**
>
> We greatly appreciate your thoughtful comments. We address your concerns as follows.
>
> ### **It would be better to show our method is generally applicable to different tasks.**
>
> As mentioned in Section 5.2 and Footnote 4, we considered other tasks (e.g., language modeling) and related models (e.g., RNN/transformer-based models), but including them in our experiments would require substantial additional implementation effort (e.g., because some PyTorch operators used in those models do not allow changing the precision of intermediate tensors). For this reason, we did not include other tasks or models. Instead, following Reviewer eNE7’s suggestion, we have added text to the abstraction and introduction (pages 1-3) to highlight that our experiments consider image classification tasks and convolutional networks.
>
> ### **A low-precision ratio may be too simple to evaluate the memory cost.**
>
> In our experiments, we measured the *average* of the low-precision ratio of a precision assignment $\pi$ over all epochs, and reported this value as the “low-precision ratio” of $\pi$ in Section 5. Hence, the reported values of “low-precision ratio” already considers the impact of our precision promotion technique on the memory cost. To clarify this point, we have added text to Sections 5.2-5.3 (pages 9-10). Furthermore, we emphasize that the impact of the precision promotion is small: the promotion consumes only a small amount of additional memory (< 3% of the maximum model aggregate), as mentioned in Sections 1 and 4.2.
>
> ### **It would be better to add an algorithm table to show our algorithm more clearly.**
>
> To make our algorithms clearer, we have added two algorithm tables to Section 4 (Algorithms 1-2 on page 7) that summarize our algorithms proposed in Sections 4.1-4.2.

---

### Review · Reviewer_E9NR · 2023-04-21

**Summary Of Contributions:**

The paper proposes a new algorithm to learn how to quantize the neural network during training with mixed floating precisions to reduce memory usage. They first theoretically analyze the memory-accuracy trade-off problem, and show that the problem is NP-Hard. To solve this problem, they propose a heuristic algorithm. Starting with a full precision mode, they gradually reduce the precision of a group of tensors (for both forward and backward) based on the parameter size and adjust them when too many overflows happen.  They empirically present that their method can reduce memory usage more than 2x over a baseline precision assignment while preserving training accuracy. Experiments are conducted on various datasets and models for image classification problems.

**Audience:**

Yes

**Broader Impact Concerns:**

I don't have any broader impact concerns.

**Claims And Evidence:**

No

**Requested Changes:**

Please view the points of weakness for details.

The two main points are listed as follows:

1) Can we make the algorithm more clear? Especially for how many rounds I need to run to reach the target quantization ratio.

2) Can we show the model can be trained in a mixed-precision way that starts with low precision?

**Strengths And Weaknesses:**

Strengths:

1) The problem is well-motivated and the paper is mostly well-written.
2) Based on the empirical results, the trade-off between memory and accuracy does exist. In addition, their method achieves a better trade-off when the model hits the compression ratio.

Weakness:

1) The algorithm is not very clear, for example, how long do I need to hit the ratio? How many iterations do I need to run when I convert one part of the model?

2) The memory usage reduction claim is vague. Is that the average usage during training or is it the one when the ratio is fixed? Also, even the ratio is fixed, it seems that during the training, the memory usage is dynamic due the promotion and re-demotion operations.

3) Starting with all high precision seems strange to me especially the main benefit of the method is memory usage reduction. With a high-precision model, the peak usage of memory is very high and limits the usage on edge devices. Then what's the point of reducing the memory?

4) It seems the proposed method works better on small models and has similar performance on larger models compared to previous methods.

5) Can we do more experiments on other tasks or larger models such as transformers?

6) The figure for ImageNet is not informative, can we zoom in a little bit at the end of the training?

---

> ### Author Response · Authors · 2023-05-02
> **Thanks for the review**
>
> We greatly appreciate your thoughtful comments. We address your concerns as follows.
>
> ### **Can you make the algorithm more clear? How many rounds do we need to run to reach the target quantization ratio?**
>
> To make our algorithms clearer, we have added two algorithm tables to Section 4 (Algorithms 1-2 on page 7) that summarize our algorithms proposed in Sections 4.1-4.2. We have also added text to Section 4 (page 6) that summarizes at a high level what these algorithms do and when they are executed.
>
> Regarding your second question, we clarify that our algorithm in Section 4.1 finds an *initial* precision assignment to be used for training and is executed *before* training starts. Given a model and a lower bound $r$ on low-precision ratio, the algorithm starts with the all-high precision assignment $\pi$ and keeps updating $\pi$ in the following way, *until* the low-precision ratio of $\pi$ is larger than $r$: iterate over the tensors of the model and demote their precision to low precision. The algorithm then returns the resulting $\pi$ as an initial precision assignment. Hence, the number of iterations taken by this algorithm depends on the model and the $r$ value, but it is at most the number of tensors of the model. Note that this algorithm does not involve any training or any forward/backward computation of the model. These explanations are given in Algorithm 1 (page 7) and surrounding text.
>
> ### **Can you show the model can be trained in a mixed-precision way that starts with low precision?**
>
> As we explained right above, our algorithm in Section 4.1 finds an initial precision assignment $\pi$ (which is mixed-precision) *before* training starts, and this $\pi$ is used from the *very beginning* of training. All our experimental results were obtained in this way. To clarify this point, we have added text to Section 4 (page 6) and edited text in Section 5.1 (page 9).
>
> ### **Is the memory usage reduction claim about the average usage during training?**
>
> Yes, our claim is about the *average* memory usage during training. We measured the average of the low-precision ratio of a precision assignment over all epochs, and reported this value in Section 5. To clarify this point, we have added text to Sections 5.2-5.3 (pages 9-10).
>
> ### **Can you do more experiments on other tasks or models such as transformers?**
>
> As mentioned in Section 5.2 and Footnote 4, we considered other tasks (e.g., language modeling) and related models (e.g., RNN/transformer-based models), but including them in our experiments would require substantial additional implementation effort (e.g., because some PyTorch operators used in those models do not allow changing the precision of intermediate tensors). For this reason, we did not include other tasks or models. Instead, following Reviewer eNE7’s suggestion, we have added text to the abstraction and introduction (pages 1-3) to highlight that our experiments consider image classification tasks and convolutional networks.
>
> ### **Can you zoom in the figure for ImageNet at the end of training?**
>
> We have added a zoomed-in version of the figure to Appendix C.1 (Figure 9 on page 23).
>
> ### **It seems the proposed method works better on small models and has similar performance on larger models, compared to previous methods.**
>
> It has been widely observed that smaller models are generally more difficult to train with low precision formats than larger models: e.g., see [Sun+2019] and [Micikevicius+2022]. Our results reaffirm this observation.

---

### Review · Reviewer_NKVw · 2023-04-25

**Summary Of Contributions:**

This work provides a technique that explores this memory-accuracy tradeoff by generating precision assignments that use less memory and at the same time lead to more accurate models, compared to the precision assignments considered by prior work in low-precision floating-point training. In one sentence, the proposed method is to assign low-precision to tensors in a decreasing order of the tensor size. Ablation studies show that this ordering is better than the increasing order and the random order. When combined with special precision promotion for those tensors encountered with overflow issues, the proposed method is shown to achieve better trade-off compared to existing methods.

**Audience:**

Yes

**Claims And Evidence:**

No

**Requested Changes:**

See the weaknesses part.

**Strengths And Weaknesses:**

Strengths:

1. The proposed method is simple enough, thus easy to implement in different hardware platforms and deep learning frameworks, which is very important for its usability.
2. The proposed method makes sense and theoretical analysis is provided.
3. Empirical performance is impressive. Extensive empirical results are provided with convincing ablation studies.

Weaknesses:

1. The authors mention several times in the paper that the precision promotion technique that handles the overflow issue can be applied to other precision assignment methods. However, related empirical results are not provided. I think this is not just a "bonus" to provide, but an essential step to prove that the proposed assignment strategy really has practical contributions standing alone. If the existing assignment methods are very competitive with the precision promotion technique, and if the overall low-precision ratio is not influenced much for sure, then the contribution of the proposed assignment strategy is undermined in my opinion. I hope the authors could provide some comparison in this perspective.

---

> ### Author Response · Authors · 2023-05-02
> **Thanks for the review**
>
> We greatly appreciate your thoughtful comments. We address your concern as follows.
>
> ### **Can you provide some comparison between our assignments and existing assignments with our precision promotion technique?**
>
> We have added this comparison to Section 5.3 (pages 10 and 12) and Appendix C.1 (text on page 22, Figures 13-15 on pages 27-28). In particular, we have run additional experiments to train models with existing assignments while applying our precision promotion technique, and have compared these results with the results from our precision assignments.
>
> In this new comparison (Figures 13-15), we have obtained the same conclusion as before: our assignments provide similar or better tradeoffs between memory and accuracy than the existing assignments, even when the latter are equipped with our precision promotion technique. We have also observed that even for existing precision assignments, our precision promotion technique successfully prevents divergence in training. These results all together show the effectiveness of our two main techniques: our precision demotion technique (Section 4.1) is effective at exploring the memory-accuracy tradeoff, and our precision promotion technique (Section 4.2) is effective at handling divergence in training.

---

### Decision · Action_Editors · 2023-06-02

**Recommendation:** Accept with minor revision

**Comment:**

Some of the reviewers still have (minor) comments relating to presentation.  The authors are requested to address these before publication.

**Audience:**

The aspect of memory efficiency is of interest to the machine learning community, particularly as models become larger and larger over time.

**Claims And Evidence:**

Reviewers had some concerns regarding gaps between some of the claims made and the extent to which they are supported.  These gaps however have largely been addressed by the authors during the discussion period.

---

> ### Author Response · Authors · 2023-06-05
> **Clarification on additional comments**
>
> We greatly appreciate the final decision on our submission! The final comment given above says: "Some of the reviewers still have (minor) comments relating to presentation. The authors are requested to address these before publication." But we cannot see any additional comments from the reviewers made after our latest response (on May 25). Could you clarify what these additional comments so that you can address them (if they are not yet addressed)?